# CO$_2$ drawdown due to particle ballasting by glacial aeolian dust: an estimate based on the ocean carbon cycle model MPIOM/HAMOCC version 1.6.2p3

Malte Heinemann[1], Joachim Segschneider[1], and Birgit Schneider[1]

[1]Kiel University, Institute of Geosciences, Ludewig-Meyn-Strasse 10, 24118 Kiel, Germany

*Correspondence to:* Malte Heinemann (malte.heinemann@ifg.uni-kiel.de)

Version: Tuesday 12[th] March, 2019, 01:14h

**Abstract.** Despite intense efforts, the mechanisms that drive glacial–interglacial changes in atmospheric pCO$_2$ are not fully understood. Here, we aim at quantifying the potential contribution of aeolian dust deposition changes to the atmospheric pCO$_2$ drawdown during the Last Glacial Maximum (LGM). To this end, we use the ocean circulation and carbon cycle model MPI-OM/HAMOCC, including a new parameterisation of particle ballasting that accounts for the acceleration of sinking organic soft tissue in the ocean by higher density biogenic calcite and opal particles, as well as mineral dust. Sensitivity experiments with reconstructed LGM dust deposition rates indicate that the acceleration of detritus by mineral dust played a small role for atmospheric pCO$_2$ variations during glacial–interglacial cycles – on the order of 5 ppmv, compared to the reconstructed ∼80 ppmv-rise of atmospheric pCO$_2$ during the last deglaciation. The additional effect of the LGM dust deposition, namely the enhanced fertilisation by the iron that is associated with the glacial dust, likely played a more important role – although the full iron fertilization effect can not be estimated in the particular model version used here due to underestimated present-day non-diazotroph iron-limitation, fertilization of diazotrophs in the tropical Pacific already leads to an atmospheric pCO$_2$ drawdown of around 10 ppmv.

## 1 Introduction

According to ice core data (Lüthi et al., 2008; Marcott et al., 2014; Köhler et al., 2017), the atmospheric CO$_2$ concentration during the Last Glacial Maximum (LGM, ∼21 thousand years ago) was about 80 ppmv lower than during the early Holocene. Model simulations show that the reduced atmospheric CO$_2$ concentration caused a global cooling, and, combined with orbital effects on Earth's climate, allowed the Laurentide, Cordilleran, British and Scandinavian ice sheets to build up (Abe-Ouchi et al., 2007; Heinemann et al., 2014; Ganopolski and Brovkin, 2017). It is still a matter of active discussion, however, which of the many potential climate and carbon cycle changes in response to the orbital forcing caused how much of the atmospheric pCO$_2$ drawdown (e.g., Brovkin et al., 2012; Chikamoto et al., 2012; Ganopolski and Brovkin, 2017).

Here we introduce a new parameterisation of the effect of higher density particles such as calcite, opal and dust on the sinking speed of detritus into the ocean circulation and carbon cycle model MPIOM/HAMOCC. Based on this new model setup, we

estimate to what extent enhanced aeolian dust deposition contributed to the reconstructed atmospheric $pCO_2$ drawdown during the last glacial period due to its effect on particle ballasting.

Observationally constrained simulations of the global dust cycle have suggested that dust production and deposition rates were at least twice as large during the LGM compared to the Holocene and present, due to 1) enhanced desert dust production, and 2) enhanced glacigenic dust production (e.g., Werner et al., 2002; Mahowald et al., 2006; Albani et al., 2016). The increased desert dust production was caused mainly by a reduction in terrestrial vegetation (leading to larger desert dust source areas) due to inverse $CO_2$-fertilisation, temperature, precipitation and cloudiness changes (Mahowald et al., 2006) or by stronger tropospheric winds (Werner et al., 2002). Glacigenic dust, which is produced by the grinding of ice sheets and glaciers over the bedrock and sediments, and the subsequent transport of the produced fine sediment by meltwater beyond the ice margin (e.g., Bullard, 2013), contributed to the enhanced dust deposition, and globally, the magnitude of this glacigenic dust contribution was comparable to that of desert dust (Mahowald et al., 2006).

It is well known that enhanced glacigenic and desert dust deposition over the oceans can strengthen the biological carbon pump due to fertilisation with the bio-available iron contained in the dust (Martin, 1990; Kumar et al., 1995; Sigman and Boyle, 2000; Werner et al., 2002). This iron hypothesis is supported for example by mesoscale iron enrichment experiments which demonstrated that primary production in one third of the world ocean is limited by iron-availability (Boyd et al., 2007). Various modelling studies testing the iron hypothesis have concluded that a range of 15-40 ppmv of the reconstructed 80 ppmv atmospheric $pCO_2$ difference can be explained by iron fertilisation due to dust deposition changes (Watson et al., 2000; Bopp et al., 2003; Brovkin et al., 2007; Hain et al., 2010; Lambert et al., 2015; Ganopolski and Brovkin, 2017).

Enhanced dust deposition can also have an effect on the sinking speed and remineralisation length scale of particulate organic carbon (POC or detritus). Detritus sinks because of its higher density compared to seawater (excess density). As aggregates including POC form, ballast minerals, i.e., silicate and calcite biominerals, as well as lithogenic minerals such as aeolian dust, which are more dense than organic tissue and seawater, provide a major weight fraction of sinking particles in the deep ocean (Honjo et al., 1982). Hence, ballast minerals contribute largely to the excess density of aggregates including detritus in the deep ocean. This led to the hypothesis that ballast minerals determine accelerated deep-water POC fluxes (ballasting hypothesis; e.g., Armstrong et al., 2002). Several deep-moored sediment trap observations of particle flux rates supported this ballasting hypothesis (see Boyd and Trull, 2007, for a review). While, e.g., Klaas and Archer (2002) and François et al. (2002) confirmed the association of calcite minerals with efficient transfer of POC to the deep ocean but suggested that the role of lithogenic minerals is negligible (see also Berelson, 2001), Dunne et al. (2007) suggested that neglecting lithogenic material in carbon cycle models would lead to an underestimation of the POC flux to the seafloor by 16-51 % globally. Noticeably high sinking speeds off Northwest Africa (Fischer and Karakaş, 2009), and artificial dust deposition experiments using natural plankton communities from off Mauritania and Saharan dust (van der Jagt et al., 2018) further support the hypothesis that dust deposition can accelerate the sinking of detritus.

Still, to the best of our knowledge, the hypothesis that particle ballasting changes due to variations of fluxes of lithogenic material into the ocean contributed to the reconstructed atmospheric $pCO_2$ changes during glacial–interglacial cycles (Ittekkot, 1993) has not yet been tested using ocean–biogeochemical models.

This manuscript is organised as follows. In Section 2, we describe our configuration of the ocean circulation and carbon cycle model MPIOM/HAMOCC (Section 2.1), introduce a simple particle ballasting scheme that includes the effect of lithogenic minerals from aeolian sources (Section 2.2), and describe the spinup procedure of the model (Section 2.3). In Section 3, we describe the necessary adjustment of HAMOCC parameters to the modified sinking speeds (Section 3.1) and compare the resulting reference simulation with the new particle ballasting to a reference simulation with the standard prescribed particle sinking speeds, and to observations (Section 3.2). In Section 4, sensitivity experiments with respect to LGM instead of modern dust deposition rates are presented to estimate the role of particle ballasting changes for galcial–interglacial atmospheric $pCO_2$ changes. Since the dust deposition fields are also used to compute the concentrations of bioavailable iron in HAMOCC, the LGM dust deposition changes have two effects, enhanced particle ballasting and iron fertilization. We isolate these two effects from each other, but will also show that the iron fertilization effect in the particular model version used in this study underestimates the observed present-day iron limitation in the Southern Ocean, likely leading to an underestimation of the iron fertilization effect of enhanced LGM dust depostion. We conclude with a summary and discussion of our results (Section 5).

## 2 Methods

To quantify the effect of aeolian dust variations on atmospheric $pCO_2$, we use the Max-Planck-Institute Ocean Model MPIOM and the embedded Hamburg Ocean Carbon Cycle model HAMOCC (Section 2.1). To account for particle ballasting effects, the default prescribed Martin-type sinking for detritus and the constant sinking speeds for opal, dust and calcite are replaced by a simple prognostic particle ballasting parameterisation following Gehlen et al. (2006), which assumes that detritus and the heavier particles form aggregates that sink at speeds computed from the density differences between these imaginary aggregates and the surrounding seawater (Section 2.2). The particle ballasting parameterisation derived here following Gehlen et al. (2006) differs for example from another approach that assumes that only a fraction of POC is sinking in association with ballast (e.g., Klaas and Archer, 2002; Armstrong et al., 2002; Howard et al., 2006; Hofmann and Schellnhuber, 2009). Note that the particle ballasting parameterization is, implicitly, also the very simplest aggregation model possible – assuming that all particulate matter instantly forms homogeneous aggregates, neglecting the complex biological and physical aggregation and disaggregation processes that occur in reality (e.g., Lam and Marchal, 2015) or that are explicitly captured in more complex (and computationally more expensive) aggregation models (e.g., Kriest and Evans, 2000).

Using the new particle ballasting parameterisation, we then perform sensitivity tests to quantify the effect of dust deposition changes on the atmosphere–ocean $CO_2$ fluxes, separating the effects of 1) particle ballasting by the dust, and 2) fertilisation by the bio-available iron associated with the dust. The presented sensitivity tests are simulations with prescribed modern or LGM dust deposition fields, atmospheric climatological forcing representing modern conditions, and pre-industrial greenhouse gas concentrations (GHGs). The atmosphere–ocean $CO_2$ flux anomalies in the simulations with LGM dust are used to estimate the effect of dust changes on atmospheric $pCO_2$ variations during glacial cycles. We do not perform transient simulations of glacial cycles yet, because 1) dust deposition fields are currently only available for snapshots during the last deglaciation, and 2) the computational cost and required wall-time to transiently simulate a glacial cycle or inception is currently too large.

## 2.1 Configuration of MPIOM/HAMOCC

Our results are based on the ocean-only model configuration of the Max-Planck-Institute Earth System Model (MPI-ESM; Giorgetta et al., 2013) version 1.2.00p4, which contains MPIOM version 1.6.2p3 (corresponding to revision 3997). MPI-ESM is a comprehensive Earth system model that consists of the atmosphere general circulation model ECHAM (Stevens et al., 2013), the land surface scheme JSBACH (Raddatz et al., 2007), and the ocean circulation model MPIOM (Jungclaus et al., 2013), which contains the ocean biogeochemistry model HAMOCC (Ilyina et al., 2013; Paulsen et al., 2017) including a sediment module (Heinze et al., 1999). MPI-ESM has been used successfully for millennial-scale atmosphere–ocean–carbon cycle simulations, including a transient simulation of the last millennium with fully interactive atmospheric $CO_2$ (Jungclaus et al., 2010). Since our aim is to isolate the effect of particle ballasting changes in the ocean during glacial–interglacial cycles on the atmosphere–ocean $CO_2$ fluxes, rather than realistically simulating past climate evolution, we use the ocean–only configuration of MPI-ESM, namely MPIOM/HAMOCC in stand-alone mode (Marsland et al., 2003; Maier-Reimer et al., 2005; Paulsen et al., 2017).

However, for the sensitivity experiments with LGM dust deposition fields, we do consider variable atmospheric $pCO_2$ with a simple atmospheric $CO_2$ box model based on atmosphere–ocean $CO_2$ flux anomalies relative to a reference simulation with modern dust deposition and ballasting (PI_BALLAST). The flux anomalies are diagnosed at the end of each simulated year, and the prescribed atmospheric $pCO_2$ is updated accordingly. For an accumulated net atmosphere–ocean $CO_2$ flux anomaly of 2.1 Gt C into the ocean, the atmospheric $CO_2$ concentration is reduced by 1 ppmv. Note that only the surface ocean biogeochemistry responds to this atmospheric $pCO_2$ feedback. There is no effect on the atmospheric radiative transfer or climate; the prescribed climatological atmospheric forcing is unchanged. If the atmospheric $CO_2$ concentration was not adjusted to the flux anomalies, a for example positive anomalous ocean $CO_2$ uptake would not result in a reduced atmospheric $pCO_2$, the atmosphere would behave like a $CO_2$ reservoir in our experiments, and the subsequent ocean $CO_2$ uptake would be overestimated (as illustrated by an additional experiment in Section 4.1 with LGM dust deposition for particle ballasting but without adjusting atmospheric $pCO_2$, shown as a blue dashed line in Fig. 5b). The long-term net ocean $CO_2$ uptake in our reference simulations (Fig. 5b) would lead to a negative $CO_2$ trend, if absolute atmosphere–ocean $CO_2$ fluxes were used; we avoid this effect by computing atmospheric $CO_2$ from the flux anomalies relative to the reference simulation in each year. Moreover, this approach of computing flux anomalies each year, compared to, e.g., the alternative of subtracting the mean flux imbalance of the reference run, avoids additional atmospheric $pCO_2$ variability due to, e.g., internal variability of the ocean circulation. By using this simple atmospheric $pCO_2$ box model in combination with MPIOM/HAMOCC rather than using the coupled MPI-ESM, atmosphere and land surface feedbacks to the $pCO_2$ variations – that would alter the ocean circulation and complicate the analysis of dust-related $CO_2$ flux anomalies – are avoided, and computational time is saved.

Another measure to save computational time is the use of a relatively coarse pre-defined, bi-polar, curvilinear ocean grid (GR30L40) with a grid-point spacing of $\sim 3°$, and 40 levels with thicknesses ranging from 10-12 m in the upper ocean to 600 m in the deepest ocean.

The ocean model MPIOM is forced by a repeating 1-year-long climatology of daily mean surface boundary conditions, including 2 m air temperature, zonal and meridional windstress, windspeed, shortwave radiation, total cloud cover, precipitation and river runoff, representing modern conditions based on the ERA40 and ERA15 reanalyses (Röske, 2005, 2006) as in Paulsen et al. (2017).

To avoid salinity trends due to imbalances of precipitation, evaporation, and river runoff, the surface salinity is restored with a relaxation coefficient of $3.3 \cdot 10^{-7}$ s$^{-1}$ to the annual salinity field of the Polar science center Hydrographic Climatology (PHC).

Note that biogeochemical changes do not affect the ocean circulation in the default (and our) configuration of MPIOM/HAMOCC. That means, for example, any effect of phytoplankton concentration changes on light absorption (Wetzel et al., 2006) is ne-
glected in MPIOM. Differences in the ocean circulation between our simulations with the standard MPIOM/HAMOCC setup and the setup that includes particle ballasting are only due to the fact that, unfortunately, different numbers of CPUs were used for the computations (Fig. 1a: black lines cover green and brown lines, but not the dark grey line of the setup without ballasting). For binary-identical results, the parameter 'nprocx' must not be changed. Otherwise, the spatial and temporal variations of the ocean circulation would be identical in the spinup with the standard model and the simulations with ballasting.

## 2.2   Particle sinking and ballasting in HAMOCC

In the standard version of HAMOCC, particle ballasting is not accounted for. Calcite and opal shells sink at a prescribed, constant speed of 30 m day$^{-1}$; dust sinks at a speed of only 0.05 m day$^{-1}$, resulting from the assumption of spherical dust particles with a diameter of 1 $\mu$m following Stokes' law (Stokes' drag is balanced by gravitational force minus buoyancy force); the sinking speed of detritus $w_{det}$ is constant in the euphotic zone, and increases with depth $z$ below the euphotic zone
($z > 100$ m) following Martin et al. (1987):

$$
w_{det}(z) = w_{eu} + \begin{cases} 0, & z \leq 100 \text{ m} \\ \frac{\lambda_{det}}{b}(z - 100 \text{ m}), & z > 100 \text{ m}, \end{cases}
\tag{1}
$$

where $w_{eu} = 3.5$ m day$^{-1}$ is the sinking speed in the euphotic zone, the parameter $b$ is set to 2, and the remineralisation rate for detritus $\lambda_{det}$ is set to 0.0025 day$^{-1}$, resulting in sinking speeds up to 80 m day$^{-1}$ (Fig. 2b).

To account for particle ballasting, we add a simple parameterisation of particle sinking speeds to HAMOCC. We assume that
all particulate matter forms aggregates and sinks together at a common speed $w_{ballast}$. As suggested by Gehlen et al. (2006), the common speed is proportional to the difference between the mean density of all particles $\rho_{particles}$ and the density of the seawater $\rho_{sw}$ in each model grid box:

$$
w_{ballast} = w_0 \cdot \frac{\rho_{particles} - \rho_{sw}}{\rho_{det} - \rho_{sw}}.
\tag{2}
$$

$w_0$ is the proportionality factor, and can be interpreted as the sinking speed of the aggregate in the absence of particle ballast.
The density difference $\rho_{particles} - \rho_{sw}$ is also called excess density. The mean density of all particles $\rho_{particles}$ is given by the mass of all particles per unit volume of seawater, divided by the volume of all particles per unit volume of seawater. In contrast

to Gehlen et al. (2006), here, dust also contributes to the ballasting of detritus:

$$\rho_{particles} = \frac{c_{dust} + \sum_b m_b \Psi_b}{\frac{1}{\rho_{dust}} c_{dust} + \sum_b \frac{m_b}{\rho_b} \Psi_b}, \tag{3}$$

where $c_{dust}$ is the mass concentration of dust (mass of dust per unit volume of seawater, in $\mathrm{g\,cm^{-3}}$), $\rho_{dust}$ is the density of dust ($2.50\ \mathrm{g\,cm^{-3}}$), $\Psi_b$ are the molar concentrations of the biogenic particle types $b$ (namely: detritus, calcite, and opal, in $\mathrm{mol\,C}$ and $\mathrm{mol\,Si}$ per $\mathrm{cm^3}$, respectively), $m_b$ are their molar masses ($32.7\ \mathrm{g\,(mol\,C)^{-1}}$ for detritus, $100.0\ \mathrm{g\,(mol\,C)^{-1}}$ for calcite, and $72.8\ \mathrm{g\,(mol\,Si)^{-1}}$ for opal) and $\rho_b$ are their densities ($1.06\ \mathrm{g\,cm^{-3}}$ for detritus, $2.71\ \mathrm{g\,cm^{-3}}$ for calcite, and $2.10\ \mathrm{g\,cm^{-3}}$ for opal).

Monthly mean dust deposition rates are prescribed based on numerical reconstructions for modern and LGM boundary conditions, including glacigenic dust (Fig. 6a-b; Mahowald et al., 2006). The dust deposition fields not only affect the sinking speed in our ballasting parameterisation, but also the availability of iron. It is assumed that 3.5 % (weight) of the dust in the water column is iron, of which 1 % is soluble and bio-available.

## 2.3 Spinup of the standard model setup

Before starting our experiments with particle ballasting, we spin-up the standard version of MPIOM/HAMOCC without particle ballasting for over 10,000 years (Fig. 1). During the spin-up phase, the ocean model, which is initialised from three-dimensional PHC temperature and salinity fields, approaches an equilibrium state (thanks to the salinity-restoring at the surface). The simulated ocean circulation is characterised by deepwater formation in the Atlantic and Southern Ocean, and no deepwater formation in the Pacific; the Atlantic meridional overturning at 26° N amounts to about 17 Sv (Sverdrup; $10^6\ \mathrm{m^3\,s^{-1}}$; Fig. 1a).

The ocean carbon cycle is not so easily tamed, since it remains an open system in our model (no restoring). To reach a stable equilibrium of the carbon cycle in the water column and the sediment, the net fluxes of organic carbon, calcite and opal from the ocean column to the sediment – which are computed in HAMOCC – need to be balanced by the fluxes into the ocean of organic matter, calcite and silicate from weathering on land – which are prescribed as namelist parameters (called deltaorg, deltacal, and deltasil). The weathering fluxes are adjusted several times during the spinup phase, aiming at 1) balancing the sediment burial fluxes as diagnosed from the trends in the sediment inventory over several hundred years prior to each weathering flux adjustment, 2) a stable calcite export production at 90 m depth of about 1 Gt C $\mathrm{yr^{-1}}$, and 3) a rain ratio (ratio of calcite export to detritus export at 90 m depth) of about 10 % (Fig. 1f). An equilibrium of the sediment, however, is not reached in our simulations. Note that the selection of the weathering fluxes itself affects the simulated biogeochemistry. For example, if the silicate weathering flux is increased, more opal can be produced, and calcite production is suppressed (see Eqs. 12-14 of Ilyina et al., 2013).

## 3 Results for modern dust deposition

To simulate a reasonably realistic carbon cycle with atmosphere–ocean $CO_2$ fluxes comparable to observations, and to create a mean state with particle ballasting that is very close to the mean state of the standard setup, it is important to adjust HAMOCC

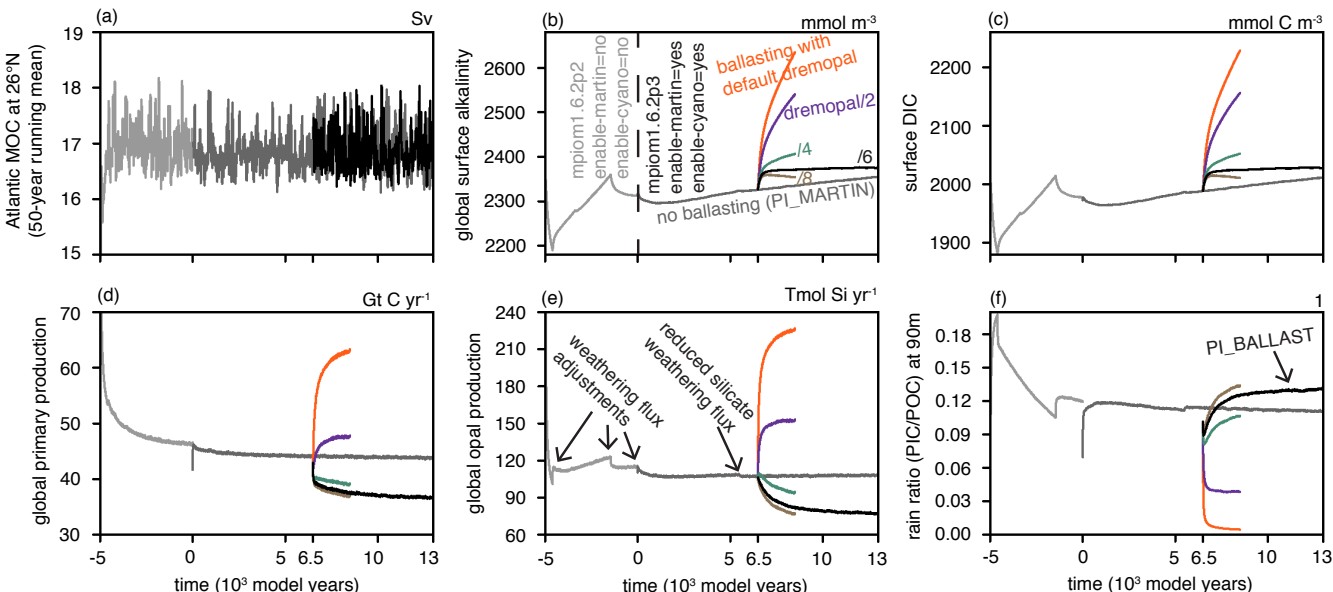

**Figure 1.** Selection of the weathering rates during the spinup of MPIOM/HAMOCC, and adjustment of the opal remineralisation rate to modified sinking speeds. a, Strength of the Atlantic Meridional Overturning Circulation (AMOC) at $26°$ N; b, surface alkalinity; c, surface dissolved inorganic carbon (DIC); d, global primary production; e, global opal production; and f, ratio of global calcite to organic carbon export at 90 m depth (rain ratio). The initial 5000-year long simulation was done with an older MPIOM/HAMOCC setup; we do not intend to interpret the biogeochemistry of this run. However, the endpoint of this older simulation is used as a starting point for our modern control simulation without particle ballasting, using the default-configuration of MPIOM/HAMOCC in MPI-ESM 1.2.00p4, including a new parameterisation of diazotrophs (Paulsen et al., 2017) and Martin-type sinking (darker grey). In model year 6500, the ballasting parameterisation is switched on, and, to compensate for the reduced sinking speed of opal in the euphotic zone, the opal dissolution rate $\lambda_{opal}$ (namelist parameter dremopal) is reduced from 0.01 day$^{-1}$ by factors of 2, 4, 6 and 8.

to the substantially modified particle sinking speeds that result from the particle ballasting parameterisation. In this section, we argue that this adjustment can be achieved by a reduction of the opal dissolution rate (Section 3.1) and then compare the resulting carbon cycle simulation with particle ballasting (PI_BALLAST) to the simulation with the standard prescribed / Martin-type POC sinking (PI_MARTIN), and to observational data (Section 3.2).

5   ## 3.1   Adjusting HAMOCC to the new particle sinking

The simulations with particle ballasting are initialized from year 6500 of the reference simulation without particle ballasting (PI_MARTIN; Fig. 1b). Using a no-ballast reference speed $w_0$ (Eq. 2) of 0.5 m day$^{-1}$ leads to global mean sinking speeds $w_{ballast}$ of the virtual aggregates between about 5 m day$^{-1}$ at the surface and about 120 m day$^{-1}$ below 5 km depth (Fig. 2b). A larger $w_0$, such as 3.5 m day$^{-1}$ as used by Gehlen et al. (2006), would lead to much higher global mean sinking speeds.

10   The advantage of using the small $w_0$ is that it yields global mean aggregate sinking speeds that are comparable to the depth-

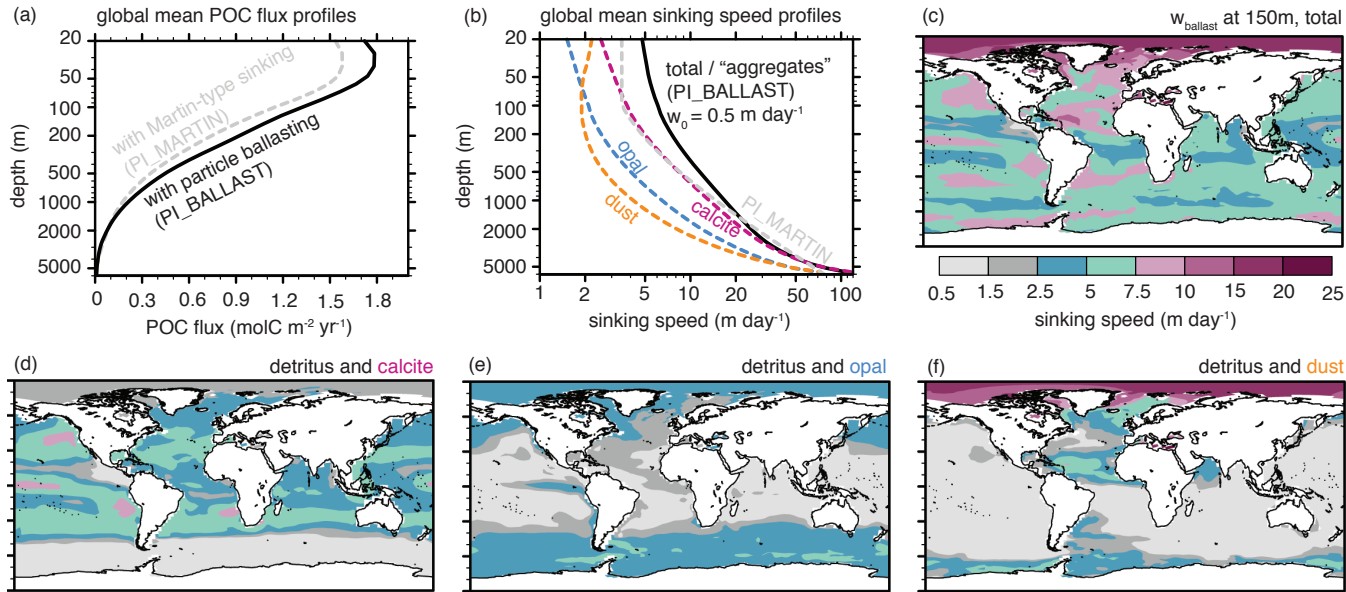

**Figure 2.** Global mean particulate organic carbon (POC) flux profiles (a) and particle sinking speed $w_{ballast}$ for b, global mean profiles and c-f, at 150 m depth. The contributions of the different ballast types (calcite, opal, and dust) to the resulting, applied "aggregate" sinking speed (black) are estimated by re-computing the sinking speed assuming that only one type of ballast is present. The dashed grey line in panel a shows the Martin-type sinking speed applied to POC in every watercolumn in the reference simulation without ballasting (PI_MARTIN). Note the logarithmic depth scale.

dependent, Martin-type sinking speed of detritus that is prescribed in PI_MARTIN (Fig. 2b), and to a similar POC flux profile (Fig. 2a), which allows us to avoid a re-tuning of the remineralisation rate of detritus.

The dissolution rate of opal, however, needs to be re-tuned, since the sinking speed of opal is reduced from previously 30 m day$^{-1}$ to now only $\sim$5 m day$^{-1}$ in the euphotic zone (Fig. 2b). Consequently, more opal is dissolved within the euphotic 5 zone, leading to higher silicate concentrations, a higher opal production rate (Fig. 1e), and a very low rain ratio (Fig. 1f) due to a completely suppressed calcite production. Without adjusting the river input (weathering) of calcite to the reduced calcite precipitation and burial, surface alkalinity and DIC rise (Fig. 1b-c). For a reduction of the opal dissolution rate by a factor of 6, the trends in alkalinity and DIC become small, and comparable to (or, actually smaller than) the trend in the control simulation without ballasting (Fig. 1b-c), and, even without re-adjusting the weathering fluxes, a stable carbon cycle simulation with 10 particle ballasting (PI_BALLAST) is achieved that is similar to PI_MARTIN.

## 3.2 Ballasting effects and comparison to observations

For pre-industrial GHGs (pCO$_2$=278 ppmv), modern ocean circulation, and modern dust deposition (Fig. 6a), the simulated atmosphere–ocean CO$_2$ fluxes in the standard setup (PI_MARTIN) and the setup with particle ballasting (PI_BALLAST) are hardly distinguishable (Fig. 3b-c). There are, however, clear differences compared to the observed modern CO$_2$ fluxes (Fig. 3a).

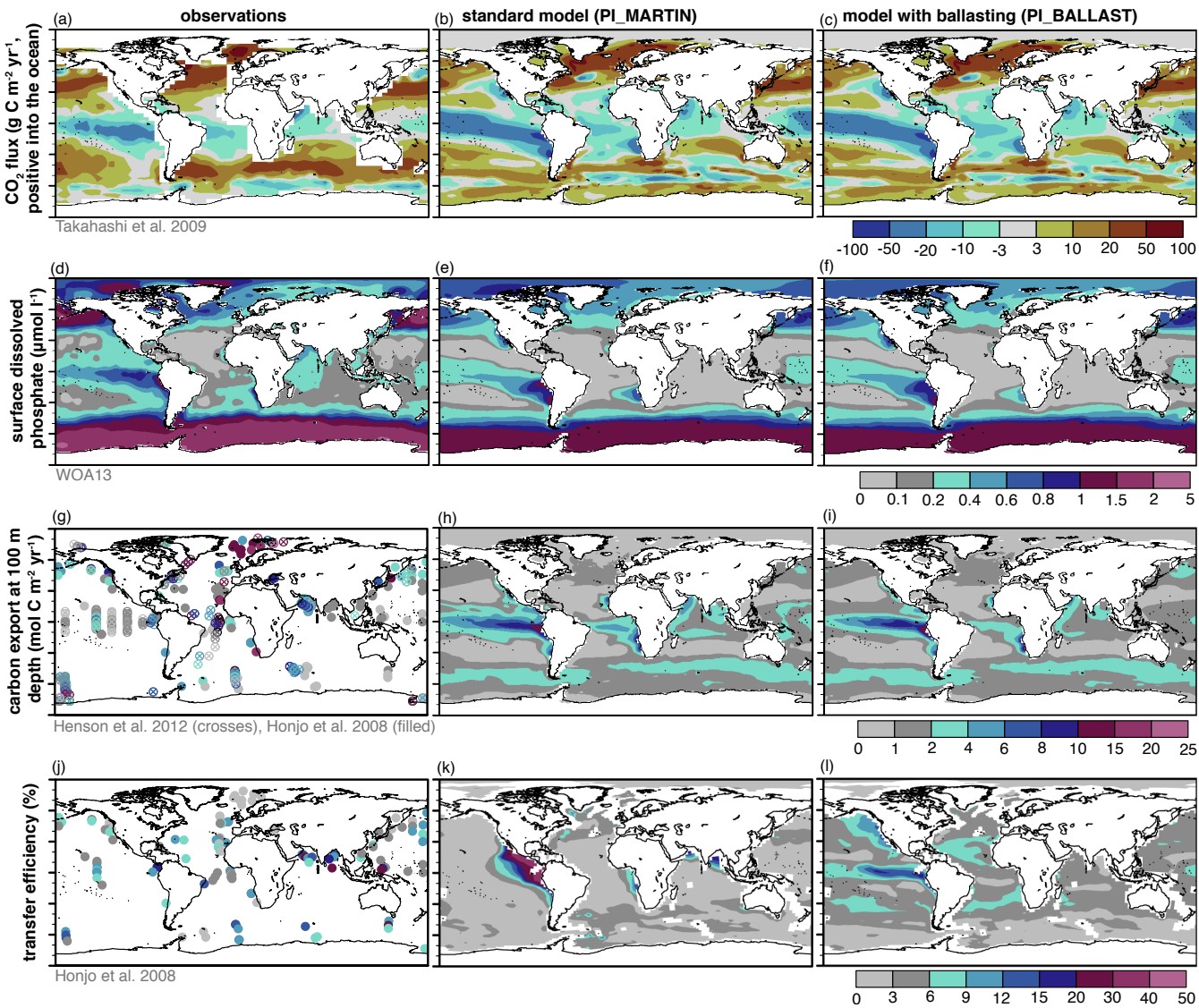

**Figure 3.** Comparison of the reference simulations with particle ballasting (PI_BALLAST) and without particle ballasting (standard setup with constant / Martin-type particle sinking; PI_MARTIN) to observations. a-c, net atmosphere-to-ocean $CO_2$ flux from observations (Takahashi et al., 2009) compared to our simulations; d-f, surface dissolved phosphate from the World Ocean Atlas (Garcia et al., 2013) compared to our simulations; g-i, carbon export at 100 m depth as derived from satellite data (Henson et al., 2012, crosses) and from sediment trap data (Honjo et al., 2008, filled circles) compared to the export at 90 m depth in our simulations; and j-l, transfer efficiency (carbon flux at 2000 m depth divided by the carbon flux at 100 m depth) as derived from sediment trap data (Honjo et al., 2008) compared to the 90 m to 2080 m transfer efficiency in our simulations. For all simulations in this figure, modern dust deposition rates were prescribed; shown are averages computed over the model years 12900 to 12999.

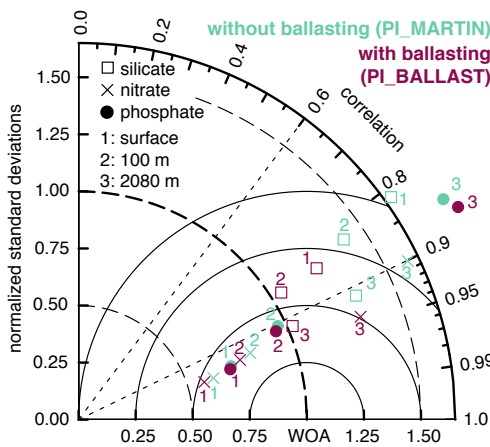

**Figure 4.** Taylor diagram comparing annual mean silicate (squares), nitrate (crosses), and phosphate concentrations (dots) at 3 different depths (numbers) of the preindustrial reference simulations with Martin-type sinking (PI_MARTIN, aquamarine) and with particle ballasting (PI_BALLAST, pink) to World Ocean Atlas data (WOA, Garcia et al., 2013).

Most notably, the observed flux of $CO_2$ from the atmosphere into the ocean between 30° S and 60° S in the South Atlantic, Indian Ocean and West Pacific is underestimated, due to an overestimation of the ocean surface $pCO_2$, as previously reported for HAMOCC within the coupled MPI-ESM (Ilyina et al., 2013). Also note that the $CO_2$ flux data (Takahashi et al., 2009) is based on modern observations, while pre-industrial GHGs are prescribed in the model simulations.

5     A comparison of simulated to observed nutrient concentrations is shown in Fig. 3d-f for phosphate at the surface, and in a Taylor diagram (Fig. 4) at the surface, 100 m depth, and 2 km depth for phosphate, nitrate, and silicate. The surface dissolved phosphate concentrations are hardly distinguishable in the simulations with particle ballasting (PI_BALLAST) and without particle ballasting (PI_MARTIN). Compared to observations (WOA, Garcia et al., 2013), the simulated phosphate concentrations in the Southern Ocean, Indian Ocean, Equatorial Pacific, and in the North Pacific are underestimated (Fig. 3d-
10  f), and these biases are again similar to previous HAMOCC studies (Ilyina et al., 2013). This similarity between the simulated phosphate concentrations is not only true at the surface, but also at depth (Fig. 4). The large overestimation of the spatial variability of the phosphate concentrations at 2 km depth in both simulations is due to an underestimation of the concentrations in the Atlantic and an overestimation in the eastern tropical Pacific (not shown). For nitrate, the observed high concentrations in the Antarctic circumpolar region at the surface and 100 m depth are underestimated in both simulations, leading to an
15  underestimated spatial variability at these depths (Fig. 4). At 2 km depth, the simulation with particle ballasting fits better to observations (Fig. 4), especially due to an improved fit in the eastern Pacific (nitrate concentrations are overestimated in the eastern Pacific in PI_MARTIN, while they fit relatively well in PI_BALLAST; not shown). For silicate concentrations, the fit to observations is improved at the surface and in 100 m depth in the simulation with particle ballasting, mostly because the concentrations in the Antarctic circumpolar region match observations well in PI_BALLAST, while they are overestimated in

PI_MARTIN. At 2 km depth, the fit is also improved in PI_BALLAST due to a reduction of the positive concentration bias in the eastern Pacific (compared to PI_MARTIN).

The global primary production in the simulation with ballasting amounts to about 39 Gt C yr$^{-1}$ compared to 44 Gt C yr$^{-1}$ in the simulation without particle ballasting (Fig. 1d), not including the primary production by diazotrophs, which amounts to

additional 5 Gt C yr$^{-1}$ and 4 Gt C yr$^{-1}$ for PI_BALLAST and PI_MARTIN, respectively. The global organic matter export is slightly reduced (7.0 Gt C yr$^{-1}$ compared to 7.5 Gt C yr$^{-1}$ without ballasting), and the global calcite production and export are only moderately larger (1 Gt C yr$^{-1}$ compared to 0.85 Gt C yr$^{-1}$ without ballasting, not shown), leading to an elevated rain ratio of about 13 % compared to 11 % in the simulation without ballasting (Fig. 1f). Opal production (Fig. 1e) and opal export at 90 m (not shown) are reduced by about 30 % in the simulation with modern dust and ballasting compared to the run

without ballasting (production 76 versus 108 Tmol Si yr$^{-1}$, export 72 versus 103 Tmol Si yr$^{-1}$).

The simulated spatial pattern of the carbon export in 90 m depth (export production) in PI_MARTIN and PI_BALLAST are very similar to each other, with the area of high export production in the Equatorial Pacific extending further into the subtropics in PI_MARTIN (Fig. 3h-i). Compared to estimates based on sediment trap data (Honjo et al., 2008) and satellite data (Henson et al., 2012), the simulated export production in both simulations is underestimated in the North Atlantic, and overestimated in

the Equatorial Pacific (Fig. 3g-i).

Only a fraction of this export production from the euphotic zone reaches the deep ocean. This fraction, also called transfer efficiency, is a measure for the efficiency of the biological pump. The range of simulated transfer efficies through the mesopelagic zone (here computed as the simulated POC flux at 2080 m depth divided by the POC flux at 90 m depth in PI_BALLAST and PI_MARTIN) compares well with the range of transfer efficiency estimates based on sediment trap data (Honjo et al., 2008,

Fig. 3j-l). In the simulation without particle ballasting (PI_MARTIN), the same sinking speed profile for POC is prescribed everywhere. Hence, the variations in the transfer efficiency are only due to local differences of the organic matter remineralization rate, which depends on oxygen availability. Denitrification and sulfate reduction remineralization rates combined are lower than aerobic remineralization rates (see Eq. 6 of Ilyina et al., 2013), leading to higher transfer efficiencies in oxygen minimum zones (OMZs), e.g., in the eastern Equatorial Pacific, Arabian Sea, and Bay of Bengal (Fig. 3k). In the simulation

with particle ballasting (PI_BALLAST), this effect of lower remineralization rates in OMZs is at least partly compensated by reduced ballasting by calcite due to the corrosive waters, resulting in lower settling speeds and transfer efficiencies (Fig. 3l). The relatively high transfer efficiencies in the mid-latitudes in PI_BALLAST compared to PI_MARTIN are in line with high particle sinking speeds mostly due to ballasting by calcite particles (compare Fig. 2d to Fig. 3l).

However, as far as we understand, even qualitatively, the global pattern of transfer efficiencies is uncertain: On the one

hand, Henson et al. (2012) suggested based on satellite data that, in large areas of the tropics and subtropics except in the eastern Equatorial Pacific cold tongue, over 30 % of the export production in 100 m depth reach the deep ocean below 2000 m, while at high latitudes (>50°) less than 10 % of the export production reaches that depth. On the other hand, Weber et al. (2016) reconstructed from nutrient distributions a "global pattern of transfer efficiency to 1000 m that is high ($\sim$25 %) at high latitudes and low ($\sim$5 %) in subtropical gyres, with intermediate values in the tropics".

Dust substantially affects sinking speeds and the transfer efficiencies in the Southern Ocean and South Atlantic off Patagonia, as well as in the equatorial Atlantic due to Saharan dust deposition, in the North Atlantic, and in the Arctic Ocean. Note, however, that, in particular in the Arctic Ocean, where POC concentrations are low, even small quantities of dust can lead to high sinking speeds, since our simple particle ballasting parameterisation only takes into account the density of the dust (in the limit of zero POC), and not its diameter. This leads to high sinking speeds in the model, while in reality, when POC concentrations are so low that particle collisions and aggregate formation are unlikely, the small and mostly individual dust particles sink very slowly according to Stoke's law. While this illustrates a limitation of the simple particle ballasting parameterisation, the effect of the erroneously high sinking speeds in the Arctic on the carbon export is small: the integrated carbon export at 90 m depth north of 80° N amounts to only 0.01 Gt C $yr^{-1}$, or just over 0.1 % of the global export of about 7 Gt C $yr^{-1}$.

In summary, although the simulated sinking speeds substantially differ between the simulations with and without particle ballasting, the biases of the setup with particle ballasting compared to observations remain similar to the biases of the standard setup. But the setup with particle ballasting now enables us to estimate the effect of glacial dust deposition on atmospheric $pCO_2$.

## 4 Sensitivity to LGM dust deposition

Starting from the simulation with particle ballasting described in the previous section (PI_BALLAST, with the opal dissolution rate reduced by a factor of 6, modern dust deposition, modern ocean circulation and pre-industrial GHGs), 3 experiments are performed to estimate the sensitivity of the ocean carbon cycle to the reconstructed LGM dust deposition. In all 3 sensitivity experiments, the prescribed monthly-mean modern dust deposition fields are replaced instantaneously by monthly-mean LGM dust deposition fields (see Fig. 5 for timeseries of the sensitivity experiments, and Fig. 6a-c for maps of the annual mean dust deposition rates and the LGM-modern dust deposition difference). In the first experiment, the LGM dust deposition rates are only used for the computation of dust concentrations in the water column, isolating the ballasting effect of the LGM dust deposition (LGM_BALLAST; solid blue lines in Fig. 5). In the second experiment, the LGM dust deposition rates are only used for the computation of the bio-available iron concentrations, isolating the iron fertilisation effect associated with the dust (LGM_IRON; orange lines). In the third experiment, the LGM dust deposition is used for both, particle ballasting and iron fertilisation (LGM_BOTH; pink lines).

### 4.1 Role of LGM dust as ballast

The largest ballasting effect of the LGM dust in our sensitivity experiments occurs in the Southern Ocean off Patagonia. Here, the higher dust deposition rates lead to about 5 to 10 m $day^{-1}$ faster particle sinking in 100 m depth (Fig. 6e), which is equivalent to about a doubling of the sinking speed (compared to Fig. 2c). This faster sinking locally leads to a higher export production (Fig. 6f), to a reduced surface dissolved inorganic carbon concentration (Fig. 6i), and consequently to a positive downward net atmosphere–ocean $CO_2$ flux anomaly (Fig. 6d), which, accumulated over the first ∼3000 years, causes an atmospheric $pCO_2$ drawdown by about 4.5 ppmv (Fig. 5c). This anomalous ocean $CO_2$ uptake occurs despite a reduction

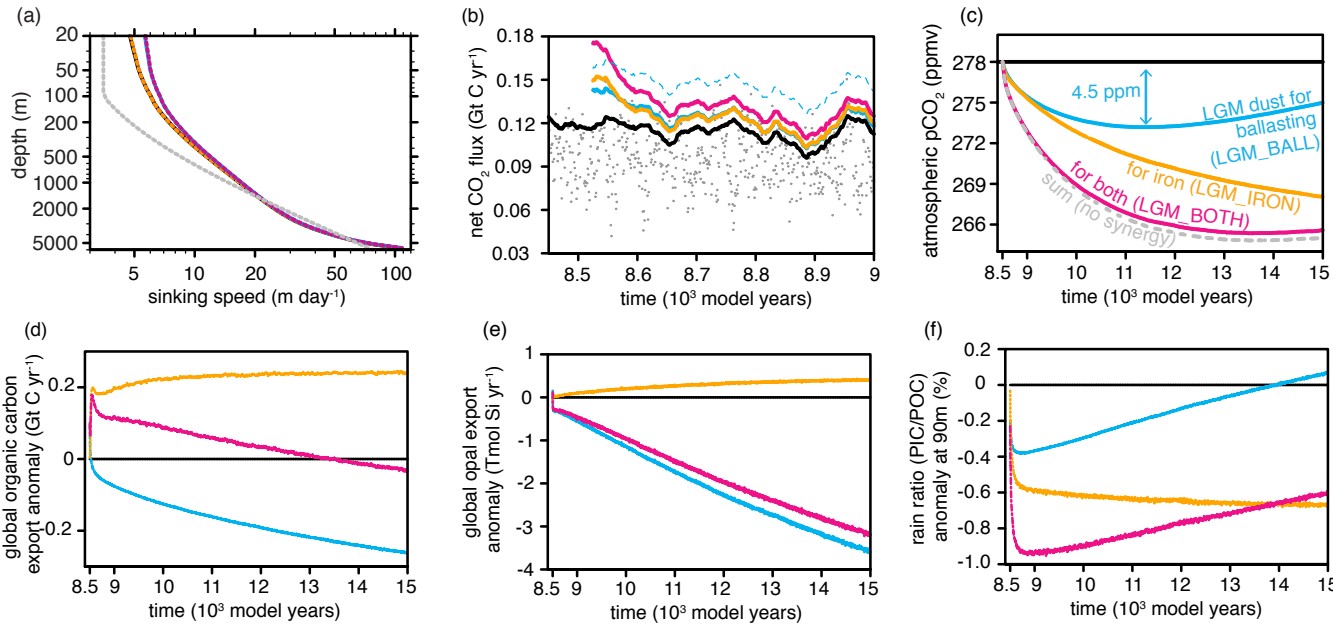

**Figure 5.** Effects of switching to LGM dust deposition for particle ballasting (LGM_BALL, blue), for the computation of iron fertilisation associated with the dust (LGM_IRON, orange), and for both (LGM_BOTH, pink) compared to the control simulation with ballasting and iron fertilisation based on modern dust deposition (PI_BALLAST, black). a, global mean sinking speed profiles; gray dashed line indicates the Martin-type sinking speed profile as used in the standard simulation without ballasting (PI_MARTIN); note that the black curve (PI_BALLAST) is hidden behind the orange curve (LGM_IRON), and that the blue curve (LGM_BALL) is hidden behind the pink curve (LGM_BOTH); b, global mean atmosphere–ocean $CO_2$ fluxes; positive values indicate a net flux from the atmosphere into the ocean; solid lines are 50-year running means; the dots show annual means of the $CO_2$ fluxes in PI_MARTIN to illustrate the large interannual variability (e.g., due to ocean circulation variability) compared to the dust-induced anomalies; the blue dashed line is the $CO_2$ flux in a simulation with LGM dust for ballasting where the atmospheric $pCO_2$ was *not* updated every year according to the atmosphere–ocean $CO_2$ flux anomalies of the previous year; c, atmospheric $pCO_2$ change based on the annual mean atmosphere–ocean $CO_2$ flux anomalies relative to PI_BALLAST; the gray dashed line shows the sum of the LGM_BALL and LGM_IRON $pCO_2$ change; d, global organic carbon export anomaly relative to PI_BALLAST in 90 m depth; e, global opal export anomaly relative to PI_BALLAST in 90 m depth; f, anomaly of the global ratio of particulate inorganic carbon (PIC, calcite) to particulate organic carbon (POC) within the particle export in 90 m depth.

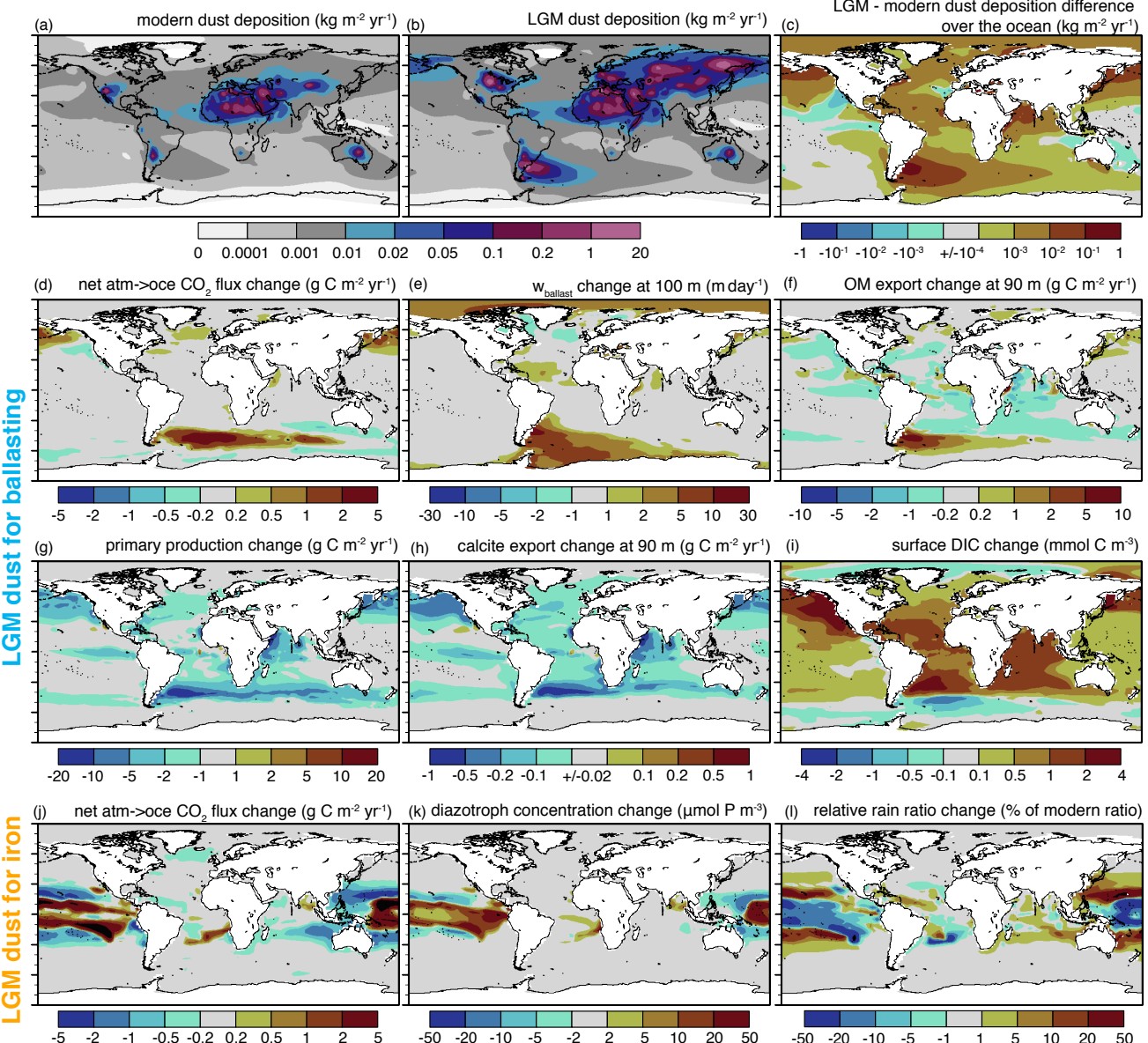

**Figure 6.** Effects of switching from modern to LGM dust deposition. a, Modern dust deposition, b, LGM dust deposition, and c, LGM-modern dust deposition difference according to Mahowald et al. (2006); d-i, LGM instead of modern dust deposition rates applied to particle ballasting (LGM_BALL) leading to changes of d, the atmosphere–ocean $CO_2$ flux (positive into the ocean), e, the common sinking speed $w_{ballast}$ in 100 m depth, f, the organic matter (OM) export in 90 m depth, g, the vertically integrated primary production, h, the calcite export in 90 m depth, and i, the surface dissolved inorganic carbon (DIC) concentration; and j-l, LGM instead of modern dust deposition rates applied to the computation of dust-related iron input (LGM_IRON) leading to changes of j, the atmosphere–ocean $CO_2$ flux, k, the concentration of diazotrophs, and l, the rain ratio (PIC/POC ratio of the export flux at 90 m depth). The changes are computed from the first 100 years of the sensitivity simulations (LGM_BALL and LGM_IRON) relative to the control simulation with modern dust deposition (PI_BALLAST).

of the globally integrated export production (Fig. 5d), as opposed to the locally enhanced export production off Patagonia (Fig. 6f). The smaller globally integrated export production is due to a reduced primary production over wide areas of the Atlantic, Pacific, and Indian Ocean, as well as parts of the Southern Ocean, just north of the area off Patagonia with enhanced organic matter export (Fig. 6g and f). This reduced primary production is consistent with a stronger nitrate depletion in surface waters (not shown) due to increased particle sinking speeds (Fig. 6e). Note that this nitrate depletion can have an effect on primary production globally, since nitrate is the least available nutrient for non-diazotroph phytoplankton everywhere in our model. However, at the same time, the production and export of calcite shells is reduced in the sensitivity simulation (Fig. 6h). Due to higher surface silicate concentrations (not shown), this calcite export reduction is disproportionately large (compared to the POC export reduction), which leads to a reduced rain ratio (Fig. 5f). This weakening of the calcium carbonate counter pump overcompensates the effect of the global reduction of the organic matter export on the atmosphere–ocean $CO_2$ fluxes, leading to a net ocean $CO_2$ uptake in the first $\sim$3000 years of our sensitivity experiment.

After about 3000 years, the ocean begins to release $CO_2$ into the atmosphere again, due to 1) the continuously decreasing export production (Fig. 5d; in response to nitrate depletion), and 2) the slowly recovering calcite production leading to an increasing PIC/POC export ratio anomaly (Fig. 5f).

The observed long-term $CO_2$ trends may also be affected by interactions with the sediment and by imbalances between sediment burial and weathering fluxes (see, e.g., Ridgwell, 2003). Since the weathering fluxes are the same in all sensitivity experiments (unchanged after the spin-up phase), a rough estimate of the role of sediment interactions can be derived from time series of the total organic carbon and calcite pools in the sediment (not shown). The organic carbon pool in the sediment in the experiment with LGM dust (LGM_BALL) grows faster than in the reference run with modern dust for ballasting (PI_BALLAST), which potentially contributes to the simulated ocean $CO_2$ uptake. This anomalous organic carbon pool trend amounts to an uptake of about 0.01 Gt C per year by the sediment, which is comparable in magnitude to the atmosphere–ocean $CO_2$ flux anomalies (Fig. 5b). Moreover, relative to PI_BALLAST, the sediment calcite pool in LGM_BALL is reduced by about $0.2 \cdot 10^{16}$ mol Ca after the first 3000 years of the sensitivity experiment, which can be translated into a global mean ocean alkalinity increase by $\sim$3 mmol m$^{-3}$, also potentially contributing to the simulated $\sim$10 mmol m$^{-3}$ alkalinty increase in LGM_BALL at the surface.

## 4.2 Role of LGM dust as a source of iron

Iron fertilisation, due to the iron contained in the dust (Section 2.2), also leads to a net $CO_2$ flux anomaly into the ocean. In our simulations, this anomalous $CO_2$ flux leads to an atmospheric $pCO_2$ drawdown by about 10 ppmv within 6500 years (Fig. 5c).

However, as evident from the lack of $CO_2$ flux anomalies in the Southern Ocean (Fig. 6j), this $CO_2$ drawdown in our sensitivity experiment is not due to the expected fertilisation of phytoplankton growth by enhanced dust deposition in the Southern Ocean. Hence, the mechanism at work in our iron sensitivity experiment differs from the frequently studied classical iron hypothesis (Martin, 1990; Kumar et al., 1995; Watson et al., 2000; Werner et al., 2002; Bopp et al., 2003; Brovkin et al., 2007; Martinez-Garcia et al., 2014; Anderson et al., 2014; Ganopolski and Brovkin, 2017). The reason for this difference is, that, in contrast to observations (see Boyd et al., 2007, for a review), non-diazotroph phytoplankton growth in our reference

simulations (for the standard setup, PI_MARTIN, and the setup with particle ballasting, PI_BALLAST) is limited by nitrate everywhere, and nowhere by iron. Since the diazotroph growth rate, unlike the non-diazotroph phytoplankton growth rate in our model, is not limited by the least available nutrient (Equation 2 of Ilyina et al., 2013), but is proportional to the product of the limiting functions for temperature, iron, phosphate and light (Equations 3 and 8 of Paulsen et al., 2017), the addition of

iron can always lead to enhanced diazotroph growth. In fact, we find enhanced diazotroph growth in particular in the tropical Pacific (Fig. 6k), which locally leads to a net positive downward atmosphere–ocean $CO_2$ flux anomaly (a reduced net $CO_2$ outgassing in the tropical Pacific; Fig. 6j). This $CO_2$ flux anomaly is caused by 1) an enhanced organic matter export due to the enhanced diazotroph production (Fig. 5d), and 2) the associated reduction of the rain ratio in the tropical Pacific leading to a weakening of the calcium carbonate counter pump (Figs. 5f and 6l). A relative shift from calcifying to silicifying organisms,

as reflected by enhanced opal export production (Fig. 5e), also contributes to the weakening of the calcium carbonate counter pump. However, the global mean sinking speed in LGM_IRON hardly differs from that in the reference run with modern dust (PI_BALLAST; Fig. 5a), suggesting that the ballasting effect of the additional opal is balanced by the effect of the reduced calcite export.

     Even after 6500 years, the ocean still takes up additional $CO_2$ from the atmosphere. We attribute most of this long-term

trend to a continuously enhanced organic matter export production (Fig. 5d) while the calcite export remains reduced, leading to a continuously lower PIC/POC ratio (Fig. 5f) and to a reduced export of alkalinity. A slightly reduced calcite pool in the sediment in LGM_IRON compared to PI_BALLAST also potentially contributes a small portion to the simulated long term atmospheric pCO$_2$ drawdown. The sediment calcite pool reduction is equivalent to an alkalinity increase in the water column by less than 2 mmol m$^{-3}$, compared to a surface alkalinity increase by about 16 mmol m$^{-3}$ after 6500 years in LGM_IRON.

Less organic carbon is stored in the sediment in LGM_IRON compared to PI_BALLAST (not shown), which means that the exchange of organic matter between the sediment and the water column does not contribute to the simulated long-term ocean $CO_2$ uptake in LGM_IRON.

## 5   Summary and discussion

We estimate the potential effect of aeolian dust deposition changes on glacial–interglacial atmospheric pCO$_2$ variations by

computing the sensitivity of the atmosphere–ocean $CO_2$ fluxes to LGM dust deposition (Section 4). To this end, we use the ocean circulation and carbon cycle model MPIOM/HAMOCC combined with a new particle ballasting parameterisation (Section 2).

     We find that the LGM dust deposition as estimated by Mahowald et al. (2006) leads to faster sinking and enhanced organic matter export off Patagonia, and to a reduction of the calcium carbonate counter pump globally, which results in an anomalous

(glacial) ocean $CO_2$ uptake that is equivalent to an atmospheric pCO$_2$ drawdown by 4.5 ppmv within 3000 years (Section 4.1).

     The iron input associated with the LGM dust deposition causes a fertilisation of diazotroph growth in the tropical Pacific, which leads to an additional atmospheric pCO$_2$ drawdown by about 10 ppmv within the 6500-year-long sensitivity experiment (Section 4.2). The combination of these effects in response to the LGM dust deposition yields an atmospheric pCO$_2$ drawdown

that is only slightly smaller than the sum of both effects (gray dashed line versus pink line in Fig. 5c), suggesting that synergistic effects play a minor role here. One negative synergistic effect, for example, which could explain the reduced $CO_2$ uptake in the combined experiment, is that the additional export production due to diazotroph iron fertilization may be reduced because nutrients are relocated due to particle ballasting.

According to previous modelling studies, iron fertilisation in the Southern Ocean played a large role, leading to an atmo-spheric $pCO_2$ drawdown during the LGM by 15-40 ppmv (Watson et al., 2000; Bopp et al., 2003; Brovkin et al., 2007; Lambert et al., 2015; Ganopolski and Brovkin, 2017). However, in our simulations, due to the abundance of iron already for the applied modern dust deposition rates, non-diazotroph phytoplankton production is nowhere iron limited, and the additional iron input associated with the enhanced LGM dust deposition does not lead to a strengthening of the biological pump in the Southern

Ocean. This lack of iron limitation could be addressed in future studies for example by reducing the prescribed constant fraction of bio-available iron within the dust in HAMOCC. It is also possible that this lack of iron limitation is due to an overestimation of the modern dust deposition rates by Mahowald et al. (2006). Both, an older dust deposition estimate by Mahowald et al. (2005) as well as the (to our knowledge) most recent dust deposition estimate by Albani et al. (2016) find (locally in the South-ern Ocean and South Atlantic as well as globally) smaller dust deposition rates than suggested by Mahowald et al. (2006).

At the HAMOCC development group at the Max-Planck-Institute for Meteorology, since MPIOM revision 4283, which is included in the recent CMIP6-release of MPI-ESM (MPI-ESM version 1.2.01, MPIOM version 1.6.3, revision 4639), the dust deposition fields by Mahowald et al. (2006) were therefore replaced by the older dust deposition product by Mahowald et al. (2005) again, leading to iron limitation in the Southern Ocean in HAMOCC (Stemmler, personal communication). However, in the model version used here, the dust by Mahowald et al. (2006) is used, and we do not swith back to the older dust de-

position field, also because this older version does not provide an estimate of the dust deposition rates during the LGM. The recent estimate by Albani et al. (2016) does provide modern as well as LGM dust deposition estimates, but this product had not yet been published at the start of this project. The dust estimate by Albani et al. (2016) will be used in future versions of MPIOM/HAMOCC, but the implementation is still ongoing work. We expect that the addition of LGM dust anomalies using a version of MPIOM/HAMOCC that is newer than revision 4283 (using dust fields by Mahowald et al., 2005) or future versions

(using dust fields by Albani et al., 2016) will lead to an additional atmospheric $pCO_2$ drawdown in response to the LGM dust deposition due to the fertilization of non-diazotroph phytoplankton in the Southern Ocean. This likely explains why the effect of the LGM iron addition on atmospheric $pCO_2$ is smaller in our simulation than suggested previously. Note that a more realistic iron fertilization response can also lead to modified synergies between iron fertilization and ballasting effects.

     Mesocosm dust deposition experiments in the Mediterranean Sea (Wagener et al., 2010), and their numerical simulation (Ye

et al., 2011), as well as more recent simulations of the global iron cycle (Ye and Völker, 2017) indicated that dust can also be a sink for dissolved bio-available iron, because the dust particles provide surfaces for adsorption of dissolved bio-available iron, leading to adsorptive scavenging. By not taking adsorptive scavenging by dust particles into account, we potentially overestimate the effect of dust deposition changes on iron fertilisation.

     Aggregate porosity and the degree of aggregate repackaging likely affects sinking speeds, too, as found for example in

mesocosm experiments (Bach et al., 2016). In particular, opal ballast during diatom blooms did not lead to higher sinking

speeds in the 25 m deep mesocosms, potentially due to the high porosities of the rather fresh aggregates. It is left for future studies to take into account the effect of porosity on the particle sinking speeds in HAMOCC. To avoid the computational expense of explicitly simulating aggregation and the porosities, the fraction of repackaged detritus (by zooplankton) to freshly produced detritus (e.g., by diatoms) could be computed and used to extend the parameterisation of particle sinking speeds.

5  Also note that no aggregate sizes, size distributions or particle shapes are being computed in our ballasting parameterization, and hence potential effects of aggregate size distribution or shape changes on sinking velocities are neglected (see, e.g., Komar et al., 1981, for sinking speeds of approximately cylindrical fecal pellets).

Particle ballasting may play a larger role for glacial–interglacial $pCO_2$ variations than suggested by the dust-sensitivity alone. Variable ratios of opal/POC or calcite/POC can also affect particle sinking speeds via ballasting, and the ratios may 10 have been modified by environmental change, such as variable temperature, variable input of weathering fluxes from land, sea-level or ocean circulation changes. For example, as discussed in Section 3.2, the efficiency of the soft tissue pump in oxygen minimum zones (OMZs) is enhanced due to reduced remineralization rates, but corrosive conditions in OMZs tend to cause reduced calcite concentrations and thereby slower particle sinking speeds, reducing the higher remineralization length scale in OMZs; i.e., the role of oxygenation changes for the soft tissue pump and atmospheric $pCO_2$ in response to solubility, 15 ocean circulation, or ecosystem variability (e.g., Jaccard and Galbraith, 2012) was potentially modified by particle ballasting. Transient simulations with the coupled MPI-ESM – including, e.g., temperature and ocean circulation changes and their effects on OMZs – are envisaged to quantify the role of these potential additional particle ballasting feedbacks for glacial–interglacial $CO_2$ variability.

*Code and data availability.*  MPIOM and HAMOCC are available to the scientific community as part of the Max-Planck-Institute Earth 20 System Model MPI-ESM (www.mpimet.mpg.de). Our results are based on MPI-ESM version 1.2.00p4, which includes MPIOM version 1.6.2p3 (corresponding to revision 3997). The modifications to the HAMOCC version 1.6.2p3 source code to include the particle ballasting parameterisation, as well as the model run script to include interactive atmospheric $pCO_2$ as described in Section 2.2 are provided as supplementary material. To build the model code used in this study, one would need to obtain MPI-ESM via the MPI-ESM-Forum and manually include the code modifications at the described locations. The data to reproduce the figures in this manuscript is available upon 25 request.

*Author contributions.*  All authors were involved in the design of the experiments. M.H. implemented the changes to MPIOM/HAMOCC, carried out the experiments, and wrote the manuscript with support from B.S. and J.S.. B.S. supervised the project.

*Competing interests.*  The authors declare that they have no conflict of interest.

*Acknowledgements.* The authors thank the Max-Planck-Institute for Meteorology for providing MPI-ESM, and Mathias Heinze, Irene Stemmler and Katharina Six for sharing their HAMOCC expertise. The authors also thank the editor Andrew Yool as well as the three anonymous referees for their constructive comments. The model simulations were performed at the German Climate Computing Centre (DKRZ). This work was supported by the German Federal Ministry of Education and Research (BMBF) as Research for Sustainable Development (FONA, www.fona.de) through the PalMod project (FKZ: 01LP1505D).

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
