# Peer review of "CO2 drawdown due to particle ballasting by glacial aeolian dust: an estimate based on the ocean carbon cycle model MPIOM/HAMOCC version 1.6.2p3"

_Geoscientific Model Development, 2018_

## Referee Comment (RC1) · Anonymous Referee #1 · 4 Sep 2018

This manuscript by Malte Heinemann et al. introduces a new parameterization of the ballasting effect in the MPI -OM/HAMOCC ocean model. This effect, in which sinking dust particles accelerate the soft tissue pump carbon export, has until now not been included in iron fertilization estimates of LGM dust. It is therefore a very welcome development. However, the convoluted (and ethically questionable) way the authors force an iron limited Southern Ocean makes the iron fertilization results very unbelievable. In addition, there is no way to estimate the robustness of the ballasting results presented here as there is no sensitivity analysis or uncertainty estimation. For these reasons I

cannot support the publication of this manuscript in its current form.

Major Comments:

The estimation of the ballasting effect was performed using only the Mahowald et al., 2005 dataset. I guess that for a theoretical study on this effect, any dust flux dataset will do, even an outdated one. But what would have happened if the authors used a different dust flux dataset, would the results have been 20 ppm pCO2 drawdown due to ballasting, or 1 ppm? To get a feel for the uncertainty of the results, the authors should either use several different (and recent!) dust flux datasets, or included a sensitivity analysis (e.g. 2x and 0.5x the Mahowald 2005 dust fluxes).

Figure 4(a): Even after 4,500 years the iron fertilization has not yet reached an equilibrium state for the atmospheric pCO2. Could you discuss that in chapter 4.4? Is there some long-term ocean feedback?

Page 15, lines 3-10: let me get this straight: Your model doesn't reproduce Southern Ocean iron fertilization using the Mahowald 2006 dust fluxes and you therefore conclude that the Mahowald 2006 dust fluxes are overestimated? And instead of including the updated version of that dataset (Albani et al., 2014), you decide to include data that you like better from an older paper from 2005, which itself is based on old model studies from 2003 and 2004? That is very sketchy. Maybe the model you are using is just bad at reproducing nutrient limitation and shouldn't be used at all for iron fertilization studies? I suggest that the authors either perform the simulations again with up-to-date estimates of dust fluxes using an updated version or a different model, or that they remove any mention of iron fertilization from the text and only discuss the ballasting effect.

Minor Comments: page 11, line 5: There are many black lines in Figure 4. Page 11, lines 30-31: The authors argue that primary production is reduced over many ocean regions because of nitrate depletion due to increased particle sinking speeds. I would add here that this is important in nitrate-limited zones. In fact, it would be interesting to

compare the relative strengths of this effect to the main ballasting effect.

**[GMDD](https://doi.org)**

---

## Referee Comment (RC2) · Anonymous Referee #2 · 12 Sep 2018

The manuscript by Heinemann et al., describes the addition of a ballasting parameterisation within the MPI-OM/HAMOCC model and is used to quantify the contribution of ballasting to glacial-interglacial changes in CO$_2$ associated with changing dust fluxes. The authors find that ballasting by dust particles has a smaller drawdown of atmospheric CO$_2$ compared with the effect of iron fertilisation when forced with glacial dust fluxes. I think this is a really interesting question to explore as there has been comparatively less focus on processes affecting organic carbon fluxes in the ocean interior than on the effects of iron fertilisation. However, I think it's difficult to reach a satisfying answer because the iron fertilisation effect in these experiments does not occur in the Southern Ocean as generally understood by the iron hypothesis. The authors are open about this in the manuscript but ultimately I think this limits the findings. I have detailed a number of comments on this as well as the ballasting parameterisation and sediment model below. If the authors are able to address this key issue then I think the manuscript would suitable for publication.

**General Comments:**

The modelled iron fertilisation effect in the model does not occur in the Southern Ocean as understood by the iron hypothesis. This has a number of issues for interpreting the results. Firstly, $CO_2$ drawdown associated with export production varies by location (DeVries et al., 2012) and therefore the $CO_2$ sensitivity for the iron fertilisation experiments may not be comparable. The sensitivity falls below the cited range in the introduction (8 ppm vs. 15-40 ppm). Secondly, changes in ballasting and sinking rates will lead to changes in nutrient distributions which could potentially enhance or reduce any export production changes associated with iron fertilisation. For example, an increase in export production with iron fertilisation may be reduced if ballasting increases sinking speeds locally relocating nutrients within the water column. For these reasons, I think the comparison of $CO_2$ changes is hard to interpret fully.

The description of the ballasting scheme, its appropriateness and impacts needs better description overall. The scheme from Gehlen et al., (2006) assigns a single sinking rate to all particle types according to the average excess density particles. While this scheme has been used previously, I think a few things need discussion: this scheme assumes a key role for particle aggregation (this is really a ballasting and aggregation parameterisation) and that this scheme differs considerably from other ballasting schemes used previously, (Howard et al., 2006; Hoffman and Schellnhuber 2009). Given the significant impact on opal sinking rates, I think this needs some thought. Additional figures, such as Taylor diagrams showing statistical fits for the new and old scheme versus observations would help assure me this scheme is working well. Please

also state all the units when describing the ballasting parameterisation.

The inclusion of sediments here is not well described or justified. The experiments don't seem to have reached a steady-state (e.g.,Figure 4a), is this because the sediments are still responding? Depending on the processes in the sediment model, there could be different responses to iron fertilisation and ballasting as ballasting will affect the ratios of particulate matter reaching the seafloor (e.g., Ridgwell 2003). Would it be possible to isolate and quantify the effect of sediments on the $CO_2$ drawdown?

**Specific Comments:**

Pg 2, lines 20 - 30: The citations for dust/lithogenic ballasting seem limited to only a few papers (Klaas and Archer 2002; Dunne et al., 2007) with a lack of more recent papers focussing on observed effects.

Pg 3, line 14: I am not sure the experiments here can be called equilibrium experiments as atmospheric $CO_2$ still seems to be changing in Figure 4a, and as also mentioned at the bottom of page 5.

Pg 3, line 33: The description of the box model of atmospheric $CO_2$ referred to here is quite limited. The description later on might be better located here.

Pg 4, lines 3-5: This is quite a lot of description of the grid-setup, does it have implications or relevance for the interpretation of the results?

Figure 2: It might be helpful to also see the global flux profile, e.g., a Martin Curve equivalent, to get a handle on how the sinking speeds contribute to changes in particulate fluxes.

Pg 7, lines 9-10: a change in the sinking rate for opal from 30 m day$^{-1}$ to 5 m day$^{-1}$ is quite dramatic. I would like some discussion about this change, e.g., how does it compare to values in literature and other models? Is this scheme better because of the explicit use of density or are there other things missing? Adding some summary plots about different tracers (see general comments) would also help clarify the impact

of this change.

Pg 8, lines 15-20: no quantification of opal export here

Pg 8, lines 26-29: As I understand, the sediment trap data presented in Honjo et al., (2008) is normalised to 2000 m using the Martin curve on the basis that gravitational settling is the dominant process at this depth. The data here is reported at 1000 m. Did you apply the same normalisation and if so can the same assumptions apply at this depth?

Figure 3: What causes the transfer efficiency pattern in the standard model (panel k)? From the previous description, it seems like this should be globally uniform.

Pg 10, line 6: I think the comparison between the ballast scheme here and Weber et al., (2016) is unwarranted as this is not the focus of the manuscript. The Weber analysis derives from an inversion of nutrient distributions and so represents the net effect of any number of potential processes. Any differences might therefore reflect the importance of other processes other than ballasting in some regions.

Pg 11, line 7: I am unfamiliar with this approach to modelling atmospheric $CO_2$, where does 2.1 Gt C / 1 ppm relationship derive from?

Figure 4: The $CO_2$ drawdown for the iron fertilisation (8 ppm) is lower than the published range mentioned in the Introduction (15 - 40 ppm). This needs some discussion, see also general comments.

Pg 14, lines 1-3: Does the weakening of the calcite export reflect a shift towards silicifying organisms? If so, does this also have an effect on ballasting sinking rates? i.e., is there a dual effect of ballasting from dust and from opal? I think these effects are quite interesting!

**References**

DeVries et al., (2012) The sequestration efficiency of the biological pump. Geophysical

Research Letters. 39 (13)

Hofmann and Schellnhuber (2009) Oceanic acidification affects marine carbon pump and triggers extended marine oxygen holes. PNAS. 106 (9)

Howard et al., (2006) Sensitivity of ocean carbon tracer distributions to particulate organic flux parameterizations. Global Biogeochemical Cycles. 20 (3)

Ridgwell (2003) An end to the "rain ratio" reign?. Geochemistry Geophysics Geosystems. 4 (6)

---

## Referee Comment (RC3) · Anonymous Referee #3 · 20 Sep 2018

Heinemann et al. introduce a parameterization of the ballasting effect in the MPIOM/HAMOCC ocean model. This effect contributes to accelerate the export of POC (by reducing remineralization rates) and has the potential to strengthen the marine biological carbon pump, with consequence for atmospheric CO2 concentrations. Furthermore, the study investigates the consequences of enhanced Fe supply to the ocean on global export production during the last ice age (Martin hypothesis). The sensitivity experiments suggest that both effects only entail a rather limited (i.e. 12 ppmv) effect on atmospheric CO2, certainly leaning towards the lower end of available

estimates from the literature.

This contribution is certainly both stimulating and timely and will certainly be of interest to the climate science community. I have to say, however, that the conclusions are somewhat weakened by the reduced sensitivity of the model to increased Fe availability. As mentioned below (last point), I would urge the authors to reconsider the modern Fe budget, which would allow the argumentation to be more relevant and certainly more convincing.

I'm not a climate modeler and as such have mostly concentrated on commenting the paleoclimatic/biogeochemical aspects of the manuscript. My comments are listed below -

As far as I understand the model set up does not account for the T-dependency of the remineralization length scale.

General comment -

As shown by Kwon et al., 2009 (NGeo), the most important parameter accounting for enhanced sequestration of $CO_2$ into the ocean interior results from the redistribution of remineralized carbon from intermediate to bottom waters. In essence, the depth at which POC is being remineralized is not critical as long as POC respiration takes place at intermediate depths, from which nutrients and $CO_2$ can rapidly be resupplied to the fertile surface ocean, with negligible consequences for atmospheric $CO_2$ concentrations.

However, if the bulk of POC remineralization takes place in the deep ocean cell, then $CO_2$ can be sequestered away from the atmosphere for centuries to millennia. So in essence, if the ballasting effect does not allow POC to be exported to the deep ocean, then one would expect the consequences for atmospheric $pCO_2$ to be small.

I was wondering if you could come up with some sense on how generally colder temperatures characteristic of the LGM in combination with the ballasting effect would

affect atmospheric CO2 concentrations. I understand that adding T-dependent POC remineralization rates would be computationally expensive. But this aspect should at least be discussed in some more details. Maybe you could also consider adding a few sentences regarding the role of dissolved O2 concentration on remineralization rates, since intermediate waters were probably better ventilated/oxygenated during the LGM (e.g. Jaccard and Galbraith, 2012 (NGeo); Galbraith and Jaccard, 2015 (QSR)).

Detailed comment -

p. 1, l. 13 – Köhler et al., 2017 do not present any ice-core CO2 data. Please remove.

p. 2, l. 3 - . . . "enhanced aridity", is probably more adequate that "enhanced desert"

p. 2, l. 3-4 - please add appropriate references

p. 2, l. 16 – please consider citing Hain et al., 2010 (GBC)

p. 11, l. 24-25 – please note that this observation is consistent with paleoceano-graphic observations, which suggest enhanced export production in the South Atlantic during the LGM as a result of Fe-bearing dust fertilization (e.g. Kumar et al., 1995 (Nature), Martinez-Garcia et al., 2014 (Nature), Anderson et al., 2014 (Phil. Trans. R. Soc.)). Furthermore, using stable nitrogen isotopes as a proxy for the relative nitrate consumption by phytoplankton, Martinez-Garcia et al., 2014 (Nature) showed that the biological carbon pump was not only stronger but also more efficient, in line with the argument outlined here.

p. 14, l. 8-10 - As mentioned above, there is ample evidence suggesting enhanced export production in the Subarctic Zone of the Southern Ocean as a result of Fe-fertilization (see reference above), including outside of the direct influence of the Patag-onian dust plume (e.g. Lamy et al., 2014 (Science). I am somewhat surprised that the model is not able to reproduce the paleoceanographic evidence.

p. 15 – I'm a bit puzzled by the final remarks. In essence you imply that Fe concentra-tions are too high in your control run, in part to the shortcomings associated with the

study published by Mahowald et al., 2006. As a consequence, adding Fe to simulate glacial conditions will not entail much of an effect on atmospheric CO2 concentrations. This certainly weakens the conclusions of the sensitivity study. Wouldn't it thus be possible to include model runs including the downscaled modern dust input?

---

## Author Comment (AC1) · 31 Dec 2018

First, we would like to thank all three referees for the work they put into reviewing this manuscript, and for their helpful comments and suggestions.

Before responding to the individual reviewer comments, we would like to discuss an issue that all three reviewers criticized, namely, the fact that in the particular version of MPIOM including HAMOCC that we used for this study, which we were restricted to because it was the default version of MPIOM available to the MPI-ESM community

at the time we started our experiments, the growth of non-diazotroph phytoplankton is limited by nitrate everywhere, and nowhere by iron. Only cyanobacterial growth is limited by iron availability. As pointed out in the manuscript and by the reviewers, there is ample evidence for the existence of iron-limited areas in the modern ocean. Our model results with respect to iron fertilization, and in particular the presented quantitative comparison of the iron fertilization effects to the effects of particle ballasting by LGM dust on atmospheric pCO2 are therefore biased (the LGM iron fertilization effect is likely underestimated).

Despite this bias, the iron fertilization results are still consistent within the model world. And we think that the presented effect of iron fertilization on cyanobacteria, which otherwise may not have been so dominant, is still relevant. We therefore would rather not follow the suggestion of the first reviewer to remove the iron fertilization results from the manuscript. However, we do agree that the focus of this manuscript should be the description of the ballasting parameterization and the ballasting effect due to enhanced LGM dust deposition, and we will clarify this in the revised manuscript.

In response to the presented lack of iron limitation, and following the reviewers' suggestion to test the iron fertilization results using a more up-to-date dust deposition estimate, we are currently working together with the HAMOCC developers at the MPI for Meteorology on the implementation of the recent dust deposition fields by Albani et al. (2016, GRL, http://dx.doi.org/10.1002/2016GL067911). Given the lower dust deposition rates of this recent estimate especially compared to Mahowald et al. 2006 (which is the estimate we used in this study; Figure 1), we anticipate that phytoplankton growth will again be iron limited in the Southern Ocean. However, this implementation and model re-tuning takes time, and ideally will happen synchronized with the developments at the MPI for Meteorology (to allow for a comparison of our results within the BMBF PalMod project, which provides the funding for this work). Even when the model adjustments to the new dust forcing are done, we would still need to re-do not only the LGM dust sensitivity simulations, but all the presented simulations, including the model spin-ups

with and without ballasting, because the problem (lack of iron limitation) occurs in both control simulations. This will take several months, with uncertain outcome.

Hence we think that, for this technical development manuscript, it is more appropriate to stick with the simulations as they are, to focus on a better description of the ballasting parameterization and effects, including an improved comparison to observations, and to clearly discuss the bias of the iron fertilization results, but to leave the complete repetition of all simulations with a future model version to a future study.

[Figure]

**Fig. 1.** Comparison of annual mean dust deposition estimates a) by Mahowald et al. 2006 (used here), b) by Albani et al. 2016 (future MPIOM versions?), and c) by Mahowald et al. 2005.

---

## Author Comment (AC2) · 22 Jan 2019

**Response to Anonymous Referee #1**

*"This manuscript by Malte Heinemann et al. introduces a new parameterization of the ballasting effect in the MPI-OM/HAMOCC ocean model. This effect, in which sinking dust particles accelerate the soft tissue pump carbon export, has until now not been included in iron fertilization estimates of LGM dust. It is therefore a very welcome development. However, the convoluted (and ethically questionable) way the authors force an iron limited Southern Ocean makes the iron fertilization results very unbelievable."*

There seems to be a misunderstanding. We do not force the Southern Ocean to be iron-limited. Quite the contrary - the Southern Ocean in our model study is *not* iron limited because we do use the more recent Mahowald et al. dust forcing from 2006, which is the default in the model version used. The decision to return to the older dust deposition reconstruction temporarily in later HAMOCC versions to achieve a more realistic iron limitation in the Southern Ocean was taken within the HAMOCC development group at the MPI for Meteorology. We do not use these later model versions; we only wanted to clarify that, if one of the later model versions with an iron-limited Southern Ocean is used (in a hypothetical future study by ourselves or somebody else), the simulated ocean $CO_2$-uptake in response to an iron addition in the Southern Ocean will likely be larger. We will emphasize in the revised manuscript that we did not use the version with the Mahowald et al. (2005) dust deposition rates (see also response to major comments (1) and (3) below).

*In addition, there is no way to estimate the robustness of the ballasting results presented here as there is no sensitivity analysis or uncertainty estimation. For these reasons I cannot support the publication of this manuscript in its current form.*

We think that the suggested sensitivity study is beyond the scope of this technical development paper, and that it is sufficient to warn the reader about the overestimated iron availability in the Southern Ocean in our reference simulation, and to discuss the potential underestimation of the effect of enhanced LGM iron deposition in this area.

*Major Comments:*
*(1) The estimation of the ballasting effect was performed using only the Mahowald et al., 2005 dataset. I guess that for a theoretical study on this effect, any dust flux dataset will do, even an outdated one. But what would have happened if the authors used a different dust flux dataset, would the results have been 20 ppm pCO2 drawdown due to ballasting, or 1 ppm? To get a feel for the uncertainty of the results, the authors should either use several different (and recent!) dust flux datasets, or include a sensitivity analysis (e.g. 2x and 0.5x the Mahowald 2005 dust fluxes).*

We agree with the referee that, in retrospect, it would have been better to use a more recent dust deposition estimate. In fact, we are currently working on the implementation of the recent estimate by Albani et al. (2016; see Figure below). We would also like to know how different our results would be if we had used the more recent dust deposition estimate. However, testing this sensitivity would basically mean that we have to re-do not only the LGM dust sensitivity simulations, but all the presented simulations, including the model spin-ups with and without ballasting. Changing the dust deposition fields will likely require re-tuning of the cyanobacteria production and will lead to a different model setup also for the control simulation without ballasting. As discussed in our general comment to all reviewers,

we think that the repetition of our simulations with updated dust fields is therefore beyond the scope of this paper.

Simply scaling the LGM dust anomaly by a factor of 0.5 or 2 would be an easier-to-achieve sensitivity study, but the meaning of the results would be similarly questionable, since the problem of too high iron availability in the control simulation would persist.

*(2) Figure 4(a): Even after 4,500 years the iron fertilization has not yet reached an equilibrium state for the atmospheric pCO2. Could you discuss that in chapter 4.4? Is there some long-term ocean feedback?*

We extended all sensitivity simulations by another 2000 years, but even after 6500 years the ocean in the LGM iron run keeps taking up more $CO_2$ than in the reference run with modern dust/iron.

We attribute this long-term trend to a continuously reduced PIC/POC ratio of the export production relative to the reference simulation, and hence a continuously reduced export of alkalinity, while the PIC/POC ratio in the LGM ballast simulation increases again over time due reduced primary productivity in response to nitrate depletion (Fig. 2).

[Figure]

*Figure 2: Anomalies of export production at 90m depth (a), export of calcite (b), and ratio of calcite versus organic matter (PIC/POC) for the simulation with LGM dust as ballast (LGM_BALL), with LGM dust for iron fertilization (LGM_IRON), and LGM dust for both (LGM_BOTH).*

Note that long-term trends can also arise if the sediment burial fluxes of organic matter, calcite and opal are not balanced by the weathering fluxes – which we did not adjust in the sensitivity simulations. We will discuss these trends in more detail in the revised manuscript.

*(3) Page 15, lines 3-10: let me get this straight: Your model doesn't reproduce Southern Ocean iron fertilization using the Mahowald 2006 dust fluxes and you therefore conclude that the Mahowald 2006 dust fluxes are overestimated? And instead of including the updated version of that dataset (Albani et al., 2014), you decide to include data that you like better from an older paper from 2005, which itself is based on old model studies from 2003 and 2004? That is very sketchy. Maybe the model you are using is just bad at reproducing nutrient limitation and shouldn't be used at all for iron fertilization studies? I suggest that the authors either perform the simulations again with up-to-date estimates of dust fluxes using an updated version or a different model, or that they remove any mention of iron fertilization from the text and only discuss the ballasting effect.*

Again, there seems to be a misunderstanding. We did not use the dataset from 2005. We only wanted to point out that, if the older dataset was used, the Southern Ocean would again be

iron limited. The Southern Ocean in our model is not iron limited, because we used the relatively newer dust deposition product.

That said, the most recent dataset by Albani et al. (2016) looks more similar to the 2005 data than to the 2006 data (see Fig. 1 in AC1 / general comment to all reviewers).

As discussed in the general comment to all authors, we would rather not remove the iron results, because the cyanobacterial response that leads to the $CO_2$ drawdown is still at least consistent within the model; although the lack of iron fertilization in the Southern Ocean is not in line with observations.

But we will clarify in the revised version of the manuscript that the focus of this study is the introduction of the ballasting parameterization, and not the iron results.

*Minor Comments:*
*page 11, line 5: There are many black lines in Figure 4.*

We will change the text to 'black line in Fig. 4a'.

*Page 11, lines 30-31: The authors argue that primary production is reduced over many ocean regions because of nitrate depletion due to increased particle sinking speeds. I would add here that this is important in nitrate-limited zones. In fact, it would be interesting to compare the relative strengths of this effect to the main ballasting effect.*

We agree that the effect of nitrate depletion is only important in nitrate-limited zones; however, in our model, the entire surface ocean is nitrate limited (page 14, line 13). Hence, the effect can play a role everywhere. We will point this out again in the revised manuscript. If any parts of the surface ocean were limited by phosphate, then the accelerated phosphate export due to higher sinking speeds would likely also lead to a reduced primary production in those areas.

Since the "main ballasting effect" is exactly the acceleration of particles, including particulate nitrate, it is unclear to us what is meant by separating the two effects. Did the comment aim at diagnosing the effects in different locations (i.e., nitrate depleted areas only versus other areas), or at performing new simulations, e.g., somehow keeping the NPP or nitrate export constant for the LGM ballasting sensitivity experiment? But then again, the constant NPP will also affect particle ballasting. Maybe the reviewer can elaborate?

---

## Author Comment (AC4) · 22 Jan 2019

**Response to Anonymous Referee #3**

Heinemann et al. introduce a parameterization of the ballasting effect in the MPIOM/HAMOCC ocean model. This effect contributes to accelerate the export of POC (by reducing remineralization rates) and has the potential to strengthen the marine biological carbon pump, with consequence for atmospheric CO2 concentrations. Furthermore, the study investigates the consequences of enhanced Fe supply to the ocean on global export production during the last ice age (Martin hypothesis). The sensitivity experiments suggest that both effects only entail a rather limited (i.e. 12 ppmv) effect on atmospheric CO2, certainly leaning towards the lower end of available estimates from the literature.

This contribution is certainly both stimulating and timely and will certainly be of interest to the climate science community. I have to say, however, that the conclusions are somewhat weakened by the reduced sensitivity of the model to increased Fe availability. As mentioned below (last point), I would urge the authors to reconsider the modern Fe budget, which would allow the argumentation to be more relevant and certainly more convincing.

*I'm not a climate modeler and as such have mostly concentrated on commenting the paleoclimatic/biogeochemical aspects of the manuscript. My comments are listed below.*

As far as I understand the model set up does not account for the T-dependency of the remineralization length scale.

**General comment**

As shown by Kwon et al., 2009 (NGeo), the most important parameter accounting for enhanced sequestration of CO2 into the ocean interior results from the redistribution of remineralized carbon from intermediate to bottom waters. In essence, the depth at which POC is being remineralized is not critical as long as POC respiration takes place at intermediate depths, from which nutrients and CO2 can rapidly be resupplied to the fertile surface ocean, with negligible consequences for atmospheric CO2 concentrations.

However, if the bulk of POC remineralization takes place in the deep ocean cell, then CO2 can be sequestered away from the atmosphere for centuries to millennia. So in essence, if the ballasting effect does not allow POC to be exported to the deep ocean, then one would expect the consequences for atmospheric pCO2 to be small.

I was wondering if you could come up with some sense on how generally colder temperatures characteristic of the LGM in combination with the ballasting effect would affect atmospheric CO2 concentrations. I understand that adding T-dependent POC remineralization rates would be computationally expensive. But this aspect should at least be discussed in some more details.

As shown in Segschneider and Bendtsen (2013) for a HAMOCC global warming experiment, the inclusion of T-dependent remineralization has a more complex impact on the carbon sequestration than one would expect from a simple remineralization depth scale change (reduction for warming, increase for cooling). Compensating effects due to changes in remineralization and hence euphotic layer nutrient supply -- driving changes in primary production -- and further complication due to shifts in the ecosystem (opal vs. calcite producers) and resulting changes in surface alkalinity and hence CO2-fluxes make it it non-trivial to make any statements on the potential magnitude of including T-dependent remineralization on atmospheric CO2. Segschneider and Bendtsen were planning to perform

corresponding experiments for a glacial ocean setup, but due to some unforeseen developments this has not materialized.

Maybe you could also consider adding a few sentences regarding the role of dissolved O2 concentration on remineralization rates, since intermediate waters were probably better ventilated/oxygenated during the LGM (e.g. Jaccard and Galbraith, 2012 (NGeo); Galbraith and Jaccard, 2015 (QSR)).

We would prefer to address this point later, when we actually have the glacial ocean set-up, rather than to speculate here. But we can add a brief statement to the discussion that one should keep this in mind.

**Detailed comment**

p. 1, l. 13 – Köhler et al., 2017 do not present any ice-core CO2 data. Please remove.

The 80ppm pCO2 difference between the early Holocene and the LGM was estimated from the CO2 data spline presented in Fig. 1a of Köhler et al. (2017). For that time period, the spline is based on data from the WAIS Divide Ice Core; we will add the reference pointing directly to this data in the revised manuscript (Marcott et al. 2014).

*p. 2, l. 3 - . . . "enhanced aridity", is probably more adequate that "enhanced desert"*

We will clarify: "... enhanced desert dust production and enhanced glacigenic dust production."

**p. 2, l. 3-4 - please add appropriate references**

We will clarify that these are also results of the modelling studies referred to in the previous sentence (in particular, Mahowald et al., 2006).

p. 2, l. 16 – please consider citing Hain et al., 2010 (GBC)

Thanks, the reference will be added to the list.

p. 11, l. 24-25 – please note that this observation is consistent with paleoceanographic observations, which suggest enhanced export production in the South Atlantic during the LGM as a result of Fe-bearing dust fertilization (e.g. Kumar et al., 1995 (Nature), Martinez-Garcia et al., 2014 (Nature), Anderson et al., 2014 (Phil. Trans. R. Soc.)). Furthermore, using stable nitrogen isotopes as a proxy for the relative nitrate consumption by phytoplankton, Martinez-Garcia et al., 2014 (Nature) showed that the biological carbon pump was not only stronger but also more efficient, in line with the argument outlined here.

We will add those results to the discussion. Thank you for pointing them out to us.

p. 14, l. 8-10 - As mentioned above, there is ample evidence suggesting enhanced export production in the Subarctic Zone of the Southern Ocean as a result of Fe- fertilization (see reference above), including outside of the direct influence of the Patagonian dust plume (e.g. Lamy et al., 2014 (Science). I am somewhat surprised that the model is not able to reproduce the paleoceanographic evidence. Yes, we were also surprised and somewhat disappointed by that result (see response to your next comment). The disappointment turned into our motivation to fix this issue by using a more recent dust deposition field.

p. 15 - I'm a bit puzzled by the final remarks. In essence you imply that Fe concentrations are too high in your control run, in part to the shortcomings associated with the study published by Mahowald et al., 2006. As a consequence, adding Fe to simulate glacial conditions will not entail much of an effect on atmospheric CO2 concentrations. This certainly weakens the conclusions of the sensitivity study. Wouldn't it thus be possible to include model runs including the downscaled modern dust input?

Understanding this may require a bit of a historical background: When starting our model development, we were not really aiming at an investigation of the iron fertilisation effect on glacial pCO2. Due to the standard model setup, however, in which dust is a source of iron, any change in the dust input intended to estimate the ballast effect on dust driven glacial pCO2 decrease, will likewise have an effect on the amount of iron from the same dust input field. Therefore, we had to single out the effects of glacial dust on iron fertilization and enhanced settling velocities. And only then it turned out that the biological production was nowhere iron limited in the standard HAMOCC version. Likewise, we (both the authors of this study, and the model developers at MPI) were limited to the Mahowald et al. 2006 dust fields, as they were the only ones available with LGM/modern (and future) fields.

As discussed in our general response to all reviewers, we are currently working on the implementation of a more recent dust deposition reconstruction by Albani et al. (2016), which is expected to lead to iron limitation of phytoplankton growth rates in the simulated Southern Ocean, in line with modern observations. However, this development will take several months at least. And, because the lack of iron limitation occurs in both control simulations with and without ballasting and not only within the sensitivity runs, including the new dust field or using a dust field that is scaled down would require the repetition of the control simulations and of the sensitivity runs, and the release of a new standard version of HAMOCC, which we think is beyond the scope of this paper. We will clarify that the main scope of this manuscript is the description of the ballasting parameterization and the estimate of the LGM dust ballasting effect on atmospheric CO2.

**References**

Marcott, S. A., et al..: Centennial Scale Changes in the Global Carbon Cycle During the Last Deglaciation, Nature, 514, 616–619, https://doi.org/10.1038/nature13799, 2014.

Segschneider, J., and J. Bendtsen (2013), Temperature-dependent remineralization in a warming ocean increases surface  $pCO_2$  through changes in marine ecosystem composition, Global Biogeochem. Cycles, 27, 1214–1225, doi:10.1002/2013GB004684.

---

## Author Comment (AC3)

**Response to Anonymous Referee #2**

*The manuscript by Heinemann et al., describes the addition of a ballasting parameterisation within the MPI-OM/HAMOCC model and is used to quantify the contribution of ballasting to glacial-interglacial changes in CO2 associated with changing dust fluxes. The authors find that ballasting by dust particles has a smaller drawdown of atmospheric CO2 compared with the effect of iron fertilisation when forced with glacial dust fluxes. I think this is a really interesting question to explore as there has been comparatively less focus on processes affecting organic carbon fluxes in the ocean interior than on the effects of iron fertilisation. However, I think it's difficult to reach a satisfying answer because the iron fertilisation effect in these experiments does not occur in the Southern Ocean as generally understood by the iron hypothesis. The authors are open about this in the manuscript but ultimately I think this limits the findings. I have detailed a number of comments on this as well as the ballasting parameterisation and sediment model below. If the authors are able to address this key issue then I think the manuscript would suitable for publication.*

*General Comments:*

*The modelled iron fertilisation effect in the model does not occur in the Southern Ocean as understood by the iron hypothesis. This has a number of issues for interpreting the results. Firstly, CO2 drawdown associated with export production varies by location (DeVries et al., 2012) and therefore the CO2 sensitivity for the iron fertilisation experiments may not be comparable. The sensitivity falls below the cited range in the introduction (8 ppm vs. 15-40 ppm).*

We agree with the reviewer and think that the presented $CO_2$ sensitivity for the iron fertilization is not comparable to the range cited in the introduction, because non-diazotrophic phytoplankton is not iron limited anywhere in our control simulations. We still think that presenting the iron sensitivity results is interesting enough, illustrating the effect of iron fertilization on cyanobacteria.

*Secondly, changes in ballasting and sinking rates will lead to changes in nutrient distributions which could potentially enhance or reduce any export production changes associated with iron fertilisation. For example, an increase in export production with iron fertilisation may be reduced if ballasting increases sinking speeds locally relocating nutrients within the water column. For these reasons, I think the comparison of CO2 changes is hard to interpret fully.*

We will point out in the revised manuscript that, when comparing the effects of iron fertilization and ballasting, potential interactions between the two effects such as in the given example have to be kept in mind. The given example effect could potentially be one contributor to the fact that the sum of the ballasting effect and the iron fertilization effect on

CO₂ is slightly larger than their combined effect on CO$_2$ (Fig. 3).

[Figure]

*Figure 3: Sum of ballasting effect and iron fertilization effect on atmospheric pCO$_2$ (grey dashed line) compared to the fertilization effect alone (orange; LGM_IRON), the ballasting effect alone (blue; LGM_BALL), and the combined effect (pink; LGM_BOTH).*

*The description of the ballasting scheme, its appropriateness and impacts needs better description overall. The scheme from Gehlen et al., (2006) assigns a single sinking rate to all particle types according to the average excess density particles. While this scheme has been used previously, I think a few things need discussion: this scheme assumes a key role for particle aggregation (this is really a ballasting and aggregation parameterisation) and that this scheme differs considerably from other ballasting schemes used previously, (Howard et al., 2006; Hoffman and Schellnhuber 2009).*

In the revised manuscript, we will discuss the differences of our ballast parameterization as compared to the schemes / type of schemes used by Klaas and Archer (2002), Howard et al. and Hofmann and Schellnhuber. As detailed in the specific comments below, we will also discuss the lack of an explicit aggregation scheme.

*Given the significant impact on opal sinking rates, I think this needs some thought. Additional figures, such as Taylor diagrams showing statistical fits for the new and old scheme versus observations would help assure me this scheme is working well.*

We will add a Taylor diagram to the revised manuscript, showing statistical fits of nutrient concentrations, including silicate, for both schemes versus World Ocean Atlas data (Fig. 4). For silicate, the diagram illustrates that the magnitude of spatial variability of the silicate distribution in the run with ballasting is closer to observations, while the correlation with observations hardly differs.

[Figure]

*Figure 4: Taylor diagram comparing annual mean silicate (squares), nitrate (crosses), and phosphate concentrations (dots) at 3 different depths (numbers) of the preindustrial reference simulations with Martin-type sinking (MARTIN, aquamarine) and with particle ballasting (BALLAST, pink) to World Ocean Atlas data (WOA; Garcia et al. 2013).*

*Please also state all the units when describing the ballasting parameterisation.*

We will add that the mass concentration c_dust is the mass of dust per unit volume of seawater (e.g., in g per cm3 seawater), and that the molar concentrations PSI_b are given in mol C and mol Si per unit volume of seawater respectively (e.g., for PSI_detritus and PSI_calcite in mol C per cm3 seawater).

*The inclusion of sediments here is not well described or justified. The experiments don't seem to have reached a steady-state (e.g.,Figure 4a), is this because the sediments are still responding? Depending on the processes in the sediment model, there could be different responses to iron fertilisation and ballasting as ballasting will affect the ratios of particulate matter reaching the seafloor (e.g., Ridgwell 2003). Would it be possible to isolate and quantify the effect of sediments on the $CO_2$ drawdown?*

Regarding the long-term trends seen in Fig. 4a, the strongest trend in atmospheric $pCO_2$ occurs in the iron fertilization experiment, and we attribute this long-term trend to a continuously reduced PIC/POC ratio of the export production relative to the reference

simulation, and hence a continuously reduced export of alkalinity, while the PIC/POC ratio in the LGM ballast simulation increases again over time due to reduced primary productivity in response to nitrate depletion (see Fig. 2 in our response to Referee #1).

However, it is still possible that changes in the sediment are contributing to the simulated long-term trend, and we will discuss this possibility in the revised manuscript. Quantifying this contribution is difficult, because the equilibration time with the sediment is very long, and equilibrium in the sediment has hardly been reached in the presented sensitivity simulations (see, e.g., the positive trend of calcite fluxes into the sediment, Fig. 5b), although we extended all sensitivity runs by another 2000 years. We do see that, despite the enhanced export production in the iron fertilization experiment (due to the enhanced cyanobacterial growth; Fig. 4c in the manuscript), the detritus flux into the sediment in that simulation is smaller than in the reference run without the extra iron (Fig. 5a), suggesting that detritus burial is not contributing to the long-term atmospheric $CO_2$ trend in the LGM iron simulation.

The standard version of MPIOM/HAMOCC does come with the activated sediment module, which was described briefly in Section 2.3 of Ilyina et al. (2013), or more extensively by Heinze et al. (1999). If it had been easily possible, we would have preferred to first turn off interactions of the ocean column with the sediment to avoid this problem. In future studies, an offline version of the sediment module that was recently developed at the MPI for Meteorology can be used to accelerate this equilibration process (for example to achieve equilibrium for the LGM, before a transient deglaciation simulation is started).

[Figure]

*Figure 5: (a) POC and (b) PIC fluxes into the sediment for the preindustrial reference runs with Martin-type sinking (gray) and with particle ballasting (black), as well as for the LGM dust sensitivity experiments using the dust only for ballasting (blue; LGM_BALL), only for iron fertilization (orange; LGM_IRON), and for both (pink; LGM_BOTH).*

**Specific Comments:**

*Pg 2, lines 20 - 30: The citations for dust/lithogenic ballasting seem limited to only a few papers (Klaas and Archer 2002; Dunne et al., 2007) with a lack of more recent papers focussing on observed effects.*
We will add more recent references in the revised manuscript (e.g., van der Jagd et al. 2018).

*Pg 3, line 14: I am not sure the experiments here can be called equilibrium experiments as atmospheric CO2 still seems to be changing in Figure 4a, and as also mentioned at the bottom of page 5.*
Agreed. The word "equilibrium" will be omitted in the revised manuscript.

*Pg 3, line 33: The description of the box model of atmospheric $CO_2$ referred to here is quite limited. The description later on might be better located here.*
We will move parts of Section 4.2 / the general description of the box model here.

*Pg 4, lines 3-5: This is quite a lot of description of the grid-setup, does it have implications or relevance for the interpretation of the results?*
The model grid-setup needs to be at least mentioned, since several pre-defined MPIOM grid setups exist. Some model parameters are set according to the resolution – for example the primary production depends on the thicknesses of the top layers (because growth rates are computed using the insolation at the top of each box). We will shorten the description in the revised manuscript.

*Figure 2: It might be helpful to also see the global flux profile, e.g., a Martin Curve equivalent, to get a handle on how the sinking speeds contribute to changes in particulate fluxes.*
A Martin-curve-equivalent will be added in the revised manuscript, illustrating that the global fluxes are enhanced above 2000m depth by the higher mean sinking speed in the simulation with ballasting (Figure 6).

[Figure]

*Figure 6: Global mean flux profiles of particulate organic carbon for the modern control simulation with Martin-type sinking (gray dashed) and the simulation with particle ballasting (black).*

*Pg 7, lines 9-10: a change in the sinking rate for opal from 30 m day$^{-1}$ to 5 m day$^{-1}$ is quite dramatic. I would like some discussion about this change, e.g., how does it compare to values in literature and other models? Is this scheme better because of the explicit use of density or are there other things missing? Adding some summary plots about different tracers (see general comments) would also help clarify the impact of this change.*

We will discuss the advantages and potential disadvantages or improvements of the ballasting scheme in more detail in the revised manuscript. The explicit calculation of the excess density allows us to test the ballasting hypothesis. As already mentioned in the manuscript, one potential improvement would be the inclusion of aggregate porosity. The effect of particle size is also missing in our parameterization – although we do know that, according to Stokes' drag, sinking velocities tend to increase with particle size. We will also discuss in more detail that the ballasting parameterization basically assumes instant formation of aggregates with the computed density, neglecting the complex biological and physical

aggregation and disaggregation processes that occur in reality (e.g., Lam and Marchal 2015) or that are explicitly captured in more complex (and computationally more expensive) aggregation models (e.g., Kriest and Evans 2000).

We would like to emphasize that the reduction of the opal sinking speed from the prescribed value of 30m/day to about 5m/day (as opal sinking within the virtual aggregates) only occurs in the euphotic zone. The sinking speed increases with depth to about 20m/day at 1km depth, to 30m/day at 3km, and to as much as 120m/day below 5km depth (within the virtual aggregate; black curve in Fig. 2a in the manuscript).

Still, the sinking speeds are small compared to, e.g., those in the ocean biogeochemical model PISCES-v2 (Aumont et al., 2015), where the speed increases from about 50m/day at the surface to about 240m/day close to 5km depth. However, also the opal dissolution rates differ between the models, with a more complex formulation in PISCES depending on temperature and saturation states, resulting in rates up to 0.025 day$^{-1}$, which is 2.5 times faster than the standard remineralization in HAMOCC, and 15 times faster than the rate used in our simulations with ballasting.

The better fit to observations of the simulated silicate concentrations in our simulations with ballasting compared to the standard version of HAMOCC shows that the ballasting parameterization is an improvement over the standard opal sinking and remineralization parameterization (see Taylor-diagram above).

*Pg 8, lines 15-20: no quantification of opal export here*
Opal production (Fig. 1e in the manuscript) and opal export at 90m (not shown) are reduced by about 30 % in the simulation with modern dust and ballasting compared to the run without ballasting (production 76 versus 108 Tmol Si yr$^{-1}$, export 72 versus 103 Tmol Si yr$^{-1}$). We will add those numbers to the revised manuscript.

*Pg 8, lines 26-29: As I understand, the sediment trap data presented in Honjo et al., (2008) is normalised to 2000 m using the Martin curve on the basis that gravitational settling is the dominant process at this depth. The data here is reported at 1000 m. Did you apply the same normalisation and if so can the same assumptions apply at this depth?*

Indeed, we accidentally compared the data to the simulated 960m export instead of to the simulated export close to 2000m depth. We corrected our mistake (Figure 7), now comparing the transfer efficiency from Honjo et al. (export at 2000m depth divided by export at 100m depth) to the simulated transfer efficiency computed from the fluxes at 2080m and 100m depth. The simulated transfer efficiencies match the data from Honjo et al. much better now; the mistake explains why we previously overestimated the transfer efficiency.

[Figure]

*Figure 7: New Fig. 3 j-l. Transfer efficiency computed from Honjo et al. (2008, panel j), compared to the simulated transfer efficiencies in the control run with Martin-type sinking (k)*

*and the run with ballasting and modern dust deposition (l) computed as the fraction of detritus export at 2080m compared to 100m depth.*

*Figure 3: What causes the transfer efficiency pattern in the standard model (panel k)? From the previous description, it seems like this should be globally uniform.*

The pattern arises because detritus remineralization rates depend on oxygen availability. Denitrification and sulfate reduction remineralization rates combined are lower than aerobic remineralization rates (see Eq. 6 of Ilyina et al., 2013), leading to higher transfer efficiencies in oxygen minimum zones (Figure 8 and Figure 7 (new Fig. 3k in manuscript)). In the simulations with particle ballasting, this effect of lower remineralization rates in oxygen minimum zones is partly compensated by reduced ballasting by calcite due to the corrosive waters, resulting in lower settling speeds and transfer efficiencies (Figure 7 (new Fig. 3l in manuscript)).

[Figure]

standard setup, top 2km mean oxygen [mol $O_2$ m$^{-3}$]

*Figure 8: Mean oxygen concentration in the upper 2km of the water column in the modern control simulation without ballasting.*

*Pg 10, line 6: I think the comparison between the ballast scheme here and Weber et al., (2016) is unwarranted as this is not the focus of the manuscript. The Weber analysis derives from an inversion of nutrient distributions and so represents the net effect of any number of potential processes. Any differences might therefore reflect the importance of other processes other than ballasting in some regions.*

We will focus on the comparison with direct flux data in the revised manuscript.

*Pg 11, line 7: I am unfamiliar with this approach to modelling atmospheric $CO_2$, where does 2.1 Gt C / 1 ppm relationship derive from?*

The relationship is an estimate based on the mass of the atmosphere, the molar masses of $CO_2$, C and air, and the assumption that the air and $CO_2$ in the atmosphere are ideal gases.

One ppmv of atmospheric $CO_2$ is equivalent to a volume of $CO_2$ = (volume of the atmosphere * $10^{-6}$). Since the volume of a gas is given by its mass m times its molar volume divided by its molar mass M, and assuming that the molar volumes of $CO_2$ and air are the same (assuming that they are ideal gases), the mass m of $CO_2$ equivalent to 1ppmv is given by $m_{CO2\ 1ppm} = 10^{-6} * M_{CO2} * m_{atm} / M_{air}$, where $M_{air}$ is the molar mass of dry air (28.96g/mol for 78.084% nitrogen, 20.946% oxygen, 0.934% argon and 0.03% $CO_2$), $m_{atm}$ is the mass of the atmosphere ($5.15\times10^{18}$kg, e.g., Trenberth and Smith, 2005), and $M_{CO2}$ is the molar mass of

$CO_2$ (44g/mol), which yields $m_{CO2\_1ppm}\approx7.82Gt$. 7.82Gt of $CO_2$ are equivalent to $7.82Gt*M_{CO2}/M_C\approx7.82*44/12Gt\approx2.13Gt$ of carbon.

*Figure 4: The CO₂ drawdown for the iron fertilisation (8 ppm) is lower than the published range mentioned in the Introduction (15 - 40 ppm). This needs some discussion, see also general comments.*

We will further clarify in the revised manuscript that the simulated iron fertilization effect is solely due to the fertilization of cyanobacteria growth, and not comparable to the previous estimates from the literature.

*Pg 14, lines 1-3: Does the weakening of the calcite export reflect a shift towards silicifying organisms? If so, does this also have an effect on ballasting sinking rates? i.e., is there a dual effect of ballasting from dust and from opal? I think these effects are quite interesting!*

We do see a shift towards silicifying organisms in the simulation with LGM dust for iron fertilization (LGM_IRON), as reflected by reduced calcite export (see Fig. 2b in response to the first reviewer) while opal export is enhanced (Fig. 9a, below). However, the global mean sinking speed in LGM_IRON hardly differs from that in the reference run with modern dust (PI_BALLAST; Fig. 9b), suggesting that the ballasting effect of the additional opal is balanced by the effect of the reduced calcite concentration.

[Figure]

*Fig. 9: (a) Opal export anomaly at 90m depth in the LGM dust sensitivity experiments using the dust only for ballasting (blue; LGM_BALL), only for iron fertilization (orange; LGM_IRON), and for both (pink; LGM_BOTH) relative to the pre-industrial reference with ballasting and modern dust (black, PI_BALLAST), and (b) global mean sinking speed profiles for the preindustrial reference run with particle ballasting (PI_BALLAST, black), and for the LGM sensitivity runs (colors as in panel a). For comparison, the gray dashed line in (b) is the applied Martin-type sinking speed in the PI reference run without ballasting.*

**References**

*DeVries et al., (2012) The sequestration efficiency of the biological pump. Geophysical Research Letters. 39 (13)*

*Hofmann and Schellnhuber (2009) Oceanic acidification affects marine carbon pump and triggers extended marine oxygen holes. PNAS. 106 (9)*

*Howard et al., (2006) Sensitivity of ocean carbon tracer distributions to particulate organic flux parameterizations. Global Biogeochemical Cycles. 20 (3)*

*Ridgwell (2003) An end to the "rain ratio" reign?. Geochemistry Geophysics Geosystems. 4 (6)*

van der Jagt et al. (2018). The ballasting effect of Saharan dust deposition on aggregate dynamics and carbon export: Aggregation, settling, and scavenging potential of marine snow. Limnology and Oceanography, 63(3), 1386–1394. http://doi.org/10.1002/lno.10779

Trenberth, K. E., & Smith, L. (2005). The Mass of the Atmosphere: A Constraint on Global Analyses. *Journal of Climate*, *18*(6), 864–875. http://doi.org/10.1175/JCLI-3299.1

Heinze, C., Maier-Reimer, E., Winguth, A. M. E., & Archer, D. (1999). A global oceanic sediment model for long-term climate studies. Global Biogeochemical Cycles, 13(1), 221–250. http://doi.org/10.1029/98GB02812

Lam, P. J., & Marchal, O. (2015). Insights into Particle Cycling from Thorium and Particle Data. Annual Review of Marine Science, 7(1), 159–184. http://doi.org/10.1146/annurev-marine-010814-015623

Kriest, I., & Evans, G. T. (2000). A vertically resolved model for phytoplankton aggregation. Journal of Earth System Science, 109(4), 453–469. http://doi.org/10.1007/BF02708333

---

## Author Response (AR1)

**Response to Anonymous Referee #1**

*"This manuscript by Malte Heinemann et al. introduces a new parameterization of the ballasting effect in the MPI-OM/HAMOCC ocean model. This effect, in which sinking dust particles accelerate the soft tissue pump carbon export, has until now not been included in iron fertilization estimates of LGM dust. It is therefore a very welcome development. However, the convoluted (and ethically questionable) way the authors force an iron limited Southern Ocean makes the iron fertilization results very unbelievable."*

There seems to be a misunderstanding. We do not force the Southern Ocean to be iron-limited. Quite the contrary - the Southern Ocean in our model study is \*not\* iron limited because we do use the more recent Mahowald et al. dust forcing from 2006, which is the default in the model version used. The decision to return to the older dust deposition reconstruction temporarily in later HAMOCC versions to achieve a more realistic iron limitation in the Southern Ocean was taken within the HAMOCC development group at the MPI for Meteorology. We do not use these later model versions; we only wanted to clarify that, if one of the later model versions with an iron-limited Southern Ocean is used (in a hypothetical future study by ourselves or somebody else), the simulated ocean $CO_2$-uptake in response to an iron addition in the Southern Ocean will likely be larger.

We emphasize in the revised manuscript that we did not use the version with the Mahowald et al. (2005) dust deposition rates (Section 5, page 17 lines 5-28 of the revised manuscript; see also response to major comments (1) and (3) below).

*In addition, there is no way to estimate the robustness of the ballasting results presented here as there is no sensitivity analysis or uncertainty estimation. For these reasons I cannot support the publication of this manuscript in its current form.*

We think that the suggested sensitivity study is beyond the scope of this technical development paper, as detailed below in our response to major comment (1).

*Major Comments:*
*(1) The estimation of the ballasting effect was performed using only the Mahowald et al., 2005 dataset. I guess that for a theoretical study on this effect, any dust flux dataset will do, even an outdated one. But what would have happened if the authors used a different dust flux dataset, would the results have been 20 ppm pCO2 drawdown due to ballasting, or 1 ppm? To get a feel for the uncertainty of the results, the authors should either use several different (and recent!) dust flux datasets, or include a sensitivity analysis (e.g. 2x and 0.5x the Mahowald 2005 dust fluxes).*

We agree with the referee that it would be better to use a more recent dust deposition estimate. In fact, we are currently working on the implementation of the recent estimate by Albani et al. (2016), which was unfortunately not yet available at the start of this project. After this implementation, not only the LGM dust sensitivity simulations would have to be re-done, but all the presented simulations, including the model spin-ups with and without ballasting. Changing the dust deposition fields will likely require re-tuning of the cyanobacteria production, it will lead to a different model setup also for the control simulation without ballasting, and require a new model release. As discussed in our general comment to all reviewers, we think that the repetition of our simulations with updated dust fields is therefore beyond the scope of this paper.

Simply scaling the LGM dust anomaly by a factor of 0.5 or 2 would be an easier-to-achieve sensitivity study, but the meaning of the results would be similarly questionable, since the problem of too high iron availability in the control simulation would persist.

*(2) Figure 4(a): Even after 4,500 years the iron fertilization has not yet reached an equilibrium state for the atmospheric pCO2. Could you discuss that in chapter 4.4? Is there some long-term ocean feedback?*

We extended all sensitivity simulations by another 2000 years, but even after 6500 years the ocean in the LGM iron run keeps taking up more $CO_2$ than in the reference run with modern dust/iron.

We attribute this long-term trend to a continuously reduced PIC/POC ratio of the export production relative to the reference simulation, and hence a continuously reduced export of alkalinity, while the PIC/POC ratio in the LGM ballast simulation increases again over time due to reduced primary productivity in response to nitrate depletion (Fig. 2 in this response; the anomalous organic matter export flux and the rain ratio anomalies in Fig. 2a and c are now also shown as Fig. 5 d and f in the revised manuscript).

[Figure]

Figure 2: Anomalies of export production at 90m depth (a), export of calcite (b), and ratio of calcite versus organic matter (PIC/POC) for the simulation with LGM dust as ballast (LGM_BALL), with LGM dust for iron fertilization (LGM_IRON), and LGM dust for both (LGM_BOTH).

Note that long-term trends can also arise if the sediment burial fluxes of organic matter, calcite and opal are not balanced by the weathering fluxes – which we did not adjust in the sensitivity simulations.

In the revised manuscript, we added a discussion of both, the role of the PIC/POC changes for the simulated long-term trend, as well as of the potential role of sediment – water column interactions to the corresponding section (now Section 4.2; page 16 lines 14-22).

*(3) Page 15, lines 3-10: let me get this straight: Your model doesn't reproduce Southern Ocean iron fertilization using the Mahowald 2006 dust fluxes and you therefore conclude that the Mahowald 2006 dust fluxes are overestimated? And instead of including the updated version of that dataset (Albani et al., 2014), you decide to include data that you like better from an older paper from 2005, which itself is based on old model studies from 2003 and 2004? That is very sketchy. Maybe the model you are using is just bad at reproducing nutrient limitation and shouldn't be used at all for iron fertilization studies? I suggest that the authors either perform the simulations again with up-to-date estimates of dust fluxes using an updated version or a different model, or that they remove any mention of iron fertilization*

*from the text and only discuss the ballasting effect.*

Again, there seems to be a misunderstanding. We did not use the dataset from 2005. We only wanted to point out that, if the older dataset was used, the Southern Ocean would again be iron limited. The Southern Ocean in our model is not iron limited, because we used the relatively newer dust deposition product.

That said, the most recent dataset by Albani et al. (2016) looks more similar to the 2005 data than to the 2006 data (see Fig. 1 in AC1 / general comment to all reviewers).

As discussed in the general comment to all authors, we would rather not remove the iron results, because the cyanobacterial response that leads to the $CO_2$ drawdown is still at least consistent within the model; although the lack of iron fertilization in the Southern Ocean is not in line with observations.

In the revised manuscript, we clarify that the focus of this study is the introduction of the ballasting parameterization and its effect, and not the iron results. To this end, we changed the title (removed "and iron addition"), modified the abstract (now explicitly mentioning diazotroph and non-diazotroph effects in the last sentence), and explain already in the introduction that the iron effect is likely underestimated, and why we still do look at the iron effect (page 3 lines 8-12).

*Minor Comments:*
*page 11, line 5: There are many black lines in Figure 4.*

The sentence was removed / content of this subsection was moved to Section 2.1 (Configuration of MPIOM/HAMOCC).

*Page 11, lines 30-31: The authors argue that primary production is reduced over many ocean regions because of nitrate depletion due to increased particle sinking speeds. I would add here that this is important in nitrate-limited zones. In fact, it would be interesting to compare the relative strengths of this effect to the main ballasting effect.*

We agree that the effect of nitrate depletion is only important in nitrate-limited zones; however, in our model, the entire surface ocean is nitrate limited (old manuscript, page 14, line 13). Hence, the effect can play a role everywhere. We point this out again in the revised manuscript (page 15, lines 5-7). If any parts of the surface ocean were limited by phosphate, then the accelerated phosphate export due to higher sinking speeds would likely also lead to a reduced primary production in those areas.

Since the "main ballasting effect" is exactly the acceleration of particles, including particulate nitrate, it is unclear to us what is meant by separating the two effects. Did the comment aim at diagnosing the effects in different locations (i.e., nitrate depleted areas only versus other areas), or at performing new simulations, e.g., somehow keeping the NPP or nitrate export constant for the LGM ballasting sensitivity experiment? But then again, the constant NPP will also affect particle ballasting. Maybe the reviewer can elaborate?

**Response to Anonymous Referee #2**

*The manuscript by Heinemann et al., describes the addition of a ballasting parameterisation within the MPI-OM/HAMOCC model and is used to quantify the contribution of ballasting to glacial-interglacial changes in $CO_2$ associated with changing dust fluxes. The authors find that ballasting by dust particles has a smaller drawdown of atmospheric $CO_2$ compared with the effect of iron fertilisation when forced with glacial dust fluxes. I think this is a really interesting question to explore as there has been comparatively less focus on processes affecting organic carbon fluxes in the ocean interior than on the effects of iron fertilisation. However, I think it's difficult to reach a satisfying answer because the iron fertilisation effect in these experiments does not occur in the Southern Ocean as generally understood by the iron hypothesis. The authors are open about this in the manuscript but ultimately I think this limits the findings. I have detailed a number of comments on this as well as the ballasting parameterisation and sediment model below. If the authors are able to address this key issue then I think the manuscript would suitable for publication.*

Please see responses to the detailed comments below.

*General Comments:*

*The modelled iron fertilisation effect in the model does not occur in the Southern Ocean as understood by the iron hypothesis. This has a number of issues for interpreting the results. Firstly, $CO_2$ drawdown associated with export production varies by location (DeVries et al., 2012) and therefore the $CO_2$ sensitivity for the iron fertilisation experiments may not be comparable. The sensitivity falls below the cited range in the introduction (8 ppm vs. 15-40 ppm).*

We agree with the reviewer and clarify in the revised manuscript that the presented $CO_2$ sensitivity for the iron fertilization is not comparable to the range cited in the introduction, because non-diazotrophic phytoplankton is not iron limited anywhere in our control simulations (this is now clarified in the last sentence of the abstract, the last paragraph of the introduction, and in an extended discussion on page 17 lines 5-28 in the revised manuscript). We still think that presenting the iron sensitivity results is interesting enough, illustrating the effect of iron fertilization on cyanobacteria.

In the revised manuscript, the focus was shifted to the ballasting parameterization and effects by more clearly pointing out the iron fertilization limitations, and especially by no longer mentioning the iron effects in the title of the manuscript.

*Secondly, changes in ballasting and sinking rates will lead to changes in nutrient distributions which could potentially enhance or reduce any export production changes associated with iron fertilisation. For example, an increase in export production with iron fertilisation may be reduced if ballasting increases sinking speeds locally relocating nutrients within the water column. For these reasons, I think the comparison of $CO_2$ changes is hard to interpret fully.*

We point out in the revised manuscript that, when comparing the effects of iron fertilization and ballasting, potential interactions between the two effects such as in the given example have to be kept in mind. We do find, however, that these effects appear to be small in our experiments; the sum of the ballasting effect and the iron fertilization effect on $CO_2$ is only

slightly larger than their combined effect on $CO_2$ (Fig. 3 in this response / now also included as Fig. 5c in the revised manuscript). But we also acknowledge that this may be different for more realistic simulations of the iron fertilization effects in the Southern Ocean (page 17, lines 27-28 in the revised manuscript).

[Figure]

*Figure 3: Sum of ballasting effect and iron fertilization effect on atmospheric pCO₂ (grey dashed line) compared to the fertilization effect alone (orange; LGM_IRON), the ballasting effect alone (blue; LGM_BALL), and the combined effect (pink; LGM_BOTH).*

*The description of the ballasting scheme, its appropriateness and impacts needs better description overall. The scheme from Gehlen et al., (2006) assigns a single sinking rate to all particle types according to the average excess density particles. While this scheme has been used previously, I think a few things need discussion: this scheme assumes a key role for particle aggregation (this is really a ballasting and aggregation parameterisation) and that this scheme differs considerably from other ballasting schemes used previously, (Howard et al., 2006; Hoffman and Schellnhuber 2009).*

In the revised manuscript, we include the suggested references and describe the main difference of our / the Gehlen et al. scheme compared to the type of schemes used therein – namely that only a fraction of the POC flux in these schemes is associated with ballast (page 3, lines 19-21). As detailed in the response to your specific comment below (to Pg 7, lines 9-10 / discussion of advantages and disadvantages of the used ballasting scheme), we also discuss the lack of an explicit aggregation scheme (page 3, lines 21-25 in the revised manuscript).

*Given the significant impact on opal sinking rates, I think this needs some thought. Additional figures, such as Taylor diagrams showing statistical fits for the new and old scheme versus observations would help assure me this scheme is working well.*

We added a Taylor diagram to the revised manuscript, showing statistical fits of nutrient concentrations, including silicate, for both schemes versus World Ocean Atlas data (Fig. 4 here and in the revised manuscript). For silicate, the diagram illustrates that the magnitude of spatial variability of the silicate distribution in the run with ballasting is closer to observations, while the correlation with observations hardly differs (added description / comparison to revised manuscript, page 10 line 17 to page 11 line 2).

[Figure]

*Figure 4: Taylor diagram comparing annual mean silicate (squares), nitrate (crosses), and phosphate concentrations (dots) at 3 different depths (numbers) of the preindustrial reference simulations with Martin-type sinking (MARTIN, aquamarine) and with particle ballasting (BALLAST, pink) to World Ocean Atlas data (WOA; Garcia et al. 2013).*

*Please also state all the units when describing the ballasting parameterisation.*

We added that the mass concentration c_dust is the mass of dust per unit volume of seawater (e.g., in g per cm3 seawater), and that the molar concentrations PSI_b are given in mol C and mol Si per unit volume of seawater respectively (e.g., for PSI_detritus and PSI_calcite in mol C per cm3 seawater).

*The inclusion of sediments here is not well described or justified. The experiments don't seem to have reached a steady-state (e.g.,Figure 4a), is this because the sediments are still responding? Depending on the processes in the sediment model, there could be different responses to iron fertilisation and ballasting as ballasting will affect the ratios of particulate matter reaching the seafloor (e.g., Ridgwell 2003). Would it be possible to isolate and quantify the effect of sediments on the $CO_2$ drawdown?*

The standard version of MPIOM/HAMOCC does come with the activated sediment module, which was described briefly in Section 2.3 of Ilyina et al. (2013), or more extensively by Heinze et al. (1999). The reference to the more detailed description of the sediment module (Heinze et al. 1999) was added to the revised manuscript. If it had been easily possible, we would have preferred to first turn off interactions of the ocean column with the sediment to avoid this problem. In future studies, an offline version of the sediment module that was recently developed at the MPI for Meteorology can be used to accelerate this equilibration process (for example to achieve equilibrium for the LGM, before a transient deglaciation simulation is started).

Regarding the long-term trends seen in Fig. 4a of the original manuscript, the strongest trend in atmospheric $pCO_2$ occurs in the iron fertilization experiment, and we attribute this longterm trend to a continuously reduced PIC/POC ratio of the export production relative to the reference simulation, and hence a continuously reduced export of alkalinity, while the PIC/POC ratio in the LGM ballast simulation increases again over time due to reduced primary productivity in response to nitrate depletion (see Fig. 2 in our response to Referee #1 / Fig. 5 d and f in the revised manuscript; a discussion of this trend was added on page 16, lines 14-16).

However, it is still possible that changes in the sediment are contributing to the simulated long-term trend, and we discuss this possibility in the revised manuscript. Quantifying this contribution is difficult, because the equilibration time with the sediment is very long, and equilibrium in the sediment has hardly been reached in the presented sensitivity simulations (see, e.g., the positive trend of calcite fluxes into the sediment, Fig. 5b), although we extended all sensitivity runs by another 2000 years. To estimate the potential contribution of sediment—water column interactions to the atmosphere—ocean $CO_2$ flux anomalies, we analyzed time series of the anomalous total organic carbon and calcite pools in the sediment (not shown). Changes in the anomalous total carbon and calcite pools can be translated directly into changes of the water column pool anomalies, because the prescribed weathering flux inputs are identical between all the simulations.

For LGM_BALL, we find that the organic carbon pool in the sediment grows faster than in the reference run with modern dust for ballasting (PI_BALLAST), potentially contributing to the simulated ocean CO2 uptake. This anomalous organic carbon pool trend amounts to an uptake of about 0.01 Gt C per year by the sediment, which is comparable in magnitude to the atmosphere–ocean $CO_2$ flux anomalies. Moreover, relative to PI_BALLAST, the sediment calcite pool in LGM_BALL is reduced by about $0.2 \cdot 10^{16}$ mol Ca after the first 3000 years of the sensitivity experiment, which can be translated into a global mean ocean alkalinity increase by $\sim 3$ mmol m$^{-3}$, also potentially contributing to the simulated $\sim 10$ mmol m$^{-3}$ alkalinty increase in LGM_BALL at the surface.

For LGM_IRON, a slightly reduced calcite pool in the sediment compared to PI_BALLAST may also contribute a small portion to the simulated long term atmospheric pCO2 drawdown. The sediment calcite pool reduction is equivalent to an alkalinity increase in the water column of less than 2 mmol m$^{-3}$, compared to a surface alkalinity increase by about 16 mmol m$^{-3}$ after 6500 years in LGM_IRON. Less organic carbon is stored in the sediment in LGM_IRON relative to the reference with modern dust (PI_BALLAST), meaning that the exchange of organic matter between the sediment and the water column does not contribute to the simulated long-term ocean $CO_2$ uptake in LGM_IRON.

In the revised manuscript, we added a discussion of the potential role of sediment—water column interactions for the LGM dust sensitivity experiments for ballasting (LGM_BALL) and iron fertilization (LGM_IRON) to the corresponding sections (now Section 4.1, page 15, lines 15-25, and Section 4.2, page 16 lines 14-22).

[Figure]

*Figure 5: (a) POC and (b) PIC fluxes into the sediment for the preindustrial reference runs with Martin-type sinking (gray) and with particle ballasting (black), as well as for the LGM dust sensitivity experiments using the dust only for ballasting (blue; LGM_BALL), only for iron fertilization (orange; LGM_IRON), and for both (pink; LGM_BOTH).*

**Specific Comments:**

*Pg 2, lines 20 - 30: The citations for dust/lithogenic ballasting seem limited to only a few papers (Klaas and Archer 2002; Dunne et al., 2007) with a lack of more recent papers focussing on observed effects.*
We added van der Jagt et al. 2018 to the reference list.

*Pg 3, line 14: I am not sure the experiments here can be called equilibrium experiments as atmospheric $CO_2$ still seems to be changing in Figure 4a, and as also mentioned at the bottom of page 5.*
Agreed. The word "equilibrium" was deleted.

*Pg 3, line 33: The description of the box model of atmospheric $CO_2$ referred to here is quite limited. The description later on might be better located here.*
The content of Section 4.2 was moved here (now page 4, lines 13-30).

*Pg 4, lines 3-5: This is quite a lot of description of the grid-setup, does it have implications or relevance for the interpretation of the results?*
The model grid-setup needs to be at least mentioned, since several pre-defined MPIOM grid setups exist. Some model parameters are set according to the resolution – for example the primary production depends on the thicknesses of the top layers (because growth rates are computed using the insolation at the top of each box). We shortened the description in the revised manuscript a bit / as far as we think is reasonable.

*Figure 2: It might be helpful to also see the global flux profile, e.g., a Martin Curve equivalent, to get a handle on how the sinking speeds contribute to changes in particulate fluxes.*
A Martin-curve-equivalent was added in the revised manuscript, illustrating that the global fluxes are enhanced above 2000m depth by the higher mean sinking speed in the simulation with ballasting (Fig. 6 here / Fig. 2a in the revised manuscript).

[Figure]

*Figure 6: Global mean flux profiles of particulate organic carbon for the modern control simulation with Martin-type sinking (gray dashed) and the simulation with particle ballasting (black).*

*Pg 7, lines 9-10: a change in the sinking rate for opal from 30 m day$^{-1}$ to 5 m day$^{-1}$ is quite dramatic. I would like some discussion about this change, e.g., how does it compare to values in literature and other models? Is this scheme better because of the explicit use of density or are there other things missing? Adding some summary plots about different tracers (see general comments) would also help clarify the impact of this change.*

We discuss the advantages and potential disadvantages or improvements of the ballasting scheme in more detail in the revised manuscript. The explicit calculation of the excess density allows us to test the ballasting hypothesis. As already mentioned in the original manuscript, one potential improvement would be the inclusion of aggregate porosity (in the revised manuscript, this is found in the paragraph starting on page 17 line 34).
Following your general comment above, we highlight in the revised manuscript that the ballasting parameterization is implicitly also an aggregation model (page 3, lines 21-25), which assumes instant formation of aggregates with the computed density, neglecting the complex biological and physical aggregation and disaggregation processes that occur in reality (e.g., Lam and Marchal 2015) or that are explicitly captured in more complex (and computationally more expensive) aggregation models (e.g., Kriest and Evans 2000). Moreover, no aggregate sizes, size distributions or particle shapes are being computed in our ballasting parameterization, and hence potential effects of aggregate size distribution or shape changes on sinking velocities are neglected (see, e.g., Komar et al., 1981, for sinking speeds of approximately cylindrical fecal pellets). We added this point to the discussion in the revised manuscript (page 18, lines 5-7).

We would like to emphasize that the reduction of the opal sinking speed from the prescribed value of 30m/day to about 5m/day (as opal sinking within the virtual aggregates) only occurs in the euphotic zone. The sinking speed increases with depth to about 20m/day at 1km depth, to 30m/day at 3km, and to as much as 120m/day below 5km depth (within the virtual aggregate; black curve in Fig. 2b in the revised manuscript). Still, the sinking speeds are small compared to, e.g., those in the ocean biogeochemical model PISCES-v2 (Aumont et al., 2015), where the speed increases from about 50m/day at the surface to about 240m/day close to 5km depth. However, also the opal dissolution rates differ between the models, with a more complex formulation in PISCES depending on temperature and saturation states, resulting in rates up to 0.025 day$^{-1}$, which is 2.5 times faster than the standard remineralization in HAMOCC, and 15 times faster than the rate used in our simulations with

ballasting. The better fit to observations of the simulated silicate concentrations in our simulations with ballasting compared to the standard version of HAMOCC suggests that the ballasting parameterization is an improvement over the standard opal sinking and remineralization parameterization (see Taylor-diagram above).

Opal production (Fig. 1e in the manuscript) and opal export at 90m (not shown) are reduced by about 30 % in the simulation with modern dust and ballasting compared to the run without ballasting (production 76 versus 108 Tmol Si yr$^{-1}$, export 72 versus 103 Tmol Si yr$^{-1}$). We added those numbers to the revised manuscript.

*Pg 8, lines 26-29: As I understand, the sediment trap data presented in Honjo et al., (2008) is normalised to 2000 m using the Martin curve on the basis that gravitational settling is the dominant process at this depth. The data here is reported at 1000 m. Did you apply the same normalisation and if so can the same assumptions apply at this depth?*

Indeed, we accidentally compared the data to the simulated 960m export instead of to the simulated export close to 2000m depth. We corrected our mistake (Figure 7 below), now comparing the transfer efficiency from Honjo et al. (export at 2000m depth divided by export at 100m depth) to the simulated transfer efficiency computed from the fluxes at 2080m and 100m depth. The simulated transfer efficiencies match the data from Honjo et al. much better now; the mistake explains why we previously overestimated the transfer efficiency. We corrected Fig. 3 and its caption in the revised manuscript accordingly.

[Figure]

*Figure 7: New Fig. 3 j-l. Transfer efficiency computed from Honjo et al. (2008, panel j), compared to the simulated transfer efficiencies in the control run with Martin-type sinking (k) and the run with ballasting and modern dust deposition (l) computed as the fraction of detritus export at 2080m compared to 100m depth.*

*Figure 3: What causes the transfer efficiency pattern in the standard model (panel k)? From the previous description, it seems like this should be globally uniform.*

The pattern arises because detritus remineralization rates depend on oxygen availability. Denitrification and sulfate reduction remineralization rates combined are lower than aerobic remineralization rates (see Eq. 6 of Ilyina et al., 2013), leading to higher transfer efficiencies in oxygen minimum zones (Figure 8 and Figure 7 / new Fig. 3k in the revised manuscript). In the simulations with particle ballasting, this effect of lower remineralization rates in oxygen minimum zones is partly compensated by reduced ballasting by calcite due to the corrosive waters, resulting in lower settling speeds and transfer efficiencies in the OMZs (Figure 7 / new Fig. 3l in the revised manuscript).

[Figure]

*Figure 8: Mean oxygen concentration in the upper 2km of the water column in the modern control simulation without ballasting.*

*Pg 10, line 6: I think the comparison between the ballast scheme here and Weber et al., (2016) is unwarranted as this is not the focus of the manuscript. The Weber analysis derives from an inversion of nutrient distributions and so represents the net effect of any number of potential processes. Any differences might therefore reflect the importance of other processes other than ballasting in some regions.*

We removed the direct comparison of our results to those of Weber et al., and focus on the comparison with direct flux data in the revised manuscript.

*Pg 11, line 7: I am unfamiliar with this approach to modelling atmospheric $CO_2$, where does 2.1 Gt C / 1 ppm relationship derive from?*

The relationship is an estimate based on the mass of the atmosphere, the molar masses of $CO_2$, C and air, and the assumption that the air and $CO_2$ in the atmosphere are ideal gases.

One ppmv of atmospheric $CO_2$ is equivalent to a volume of $CO_2$ = (volume of the atmosphere * $10^{-6}$). Since the volume of a gas is given by its mass m times its molar volume divided by its molar mass M, and assuming that the molar volumes of $CO_2$ and air are the same (assuming that they are ideal gases), the mass m of $CO_2$ equivalent to 1ppmv is given by $m_{CO2\_1ppm} = 10^{-6} * M_{CO2} * m_{atm} / M_{air}$, where $M_{air}$ is the molar mass of dry air (28.96g/mol for 78.084% nitrogen, 20.946% oxygen, 0.934% argon and 0.03% $CO_2$), $m_{atm}$ is the mass of the atmosphere ($5.15\times10^{18}$kg, e.g., Trenberth and Smith, 2005), and $M_{CO2}$ is the molar mass of $CO_2$ (44g/mol), which yields $m_{CO2\_1ppm} \approx 7.82$Gt. 7.82Gt of $CO_2$ are equivalent to $7.82Gt*M_{CO2}/M_C \approx 7.82*44/12Gt \approx 2.13$Gt of carbon.

*Figure 4: The $CO_2$ drawdown for the iron fertilisation (8 ppm) is lower than the published range mentioned in the Introduction (15 - 40 ppm). This needs some discussion, see also general comments.*

In the revised manuscript, we further highlight that the simulated iron fertilization effect is solely due to the fertilization of cyanobacteria growth and therefore not comparable to the previous estimates from the literature. We further clarify that the focus of the manuscript is the description of the ballasting parameterization, and the estimate of the LGM dust ballasting effect on atmospheric $CO_2$. To this end, we removed "and iron addition" from the manuscript title, modified the abstract (now explicitly mentioning diazotroph and non-diazotroph effects in the last sentence), and we explain already in the introduction that the iron effect is likely underestimated (and why we still do look at the iron effect; page 3 lines 8-

12). We point out that this underestimation likely explains the deviation from previous estimates (page 17, lines 26-27).

*Pg 14, lines 1-3: Does the weakening of the calcite export reflect a shift towards silicifying organisms? If so, does this also have an effect on ballasting sinking rates? i.e., is there a dual effect of ballasting from dust and from opal? I think these effects are quite interesting!*

We do see a shift towards silicifying organisms in the simulation with LGM dust for iron fertilization (LGM_IRON), as reflected by reduced calcite export (see Fig. 2b in this document / response to the first reviewer) while opal export is enhanced (Fig. 9a, below). However, the global mean sinking speed in LGM_IRON hardly differs from that in the reference run with modern dust (PI_BALLAST; Fig. 9b), suggesting that the ballasting effect of the additional opal is balanced by the effect of the reduced calcite concentration.

[Figure]

*Fig. 9: (a) Opal export anomaly at 90m depth in the LGM dust sensitivity experiments using the dust only for ballasting (blue; LGM_BALL), only for iron fertilization (orange; LGM_IRON), and for both (pink; LGM_BOTH) relative to the pre-industrial reference with ballasting and modern dust (black, PI_BALLAST), and (b) global mean sinking speed profiles for the preindustrial reference run with particle ballasting (PI_BALLAST, black), and for the LGM sensitivity runs (colors as in panel a). For comparison, the gray dashed line in (b) is the applied Martin-type sinking speed in the PI reference run without ballasting.*

This discussion was added to the revised manuscript (page 16, lines 9-13), and the panels a and b of Figure 9 (from this response / above) were added to the revised Fig. 5.

**Response to Anonymous Referee #3**

*Heinemann et al. introduce a parameterization of the ballasting effect in the MPIOM/HAMOCC ocean model. This effect contributes to accelerate the export of POC (by reducing remineralization rates) and has the potential to strengthen the marine biological carbon pump, with consequence for atmospheric CO2 concentrations. Furthermore, the study investigates the consequences of enhanced Fe supply to the ocean on global export production during the last ice age (Martin hypothesis). The sensitivity experiments suggest that both effects only entail a rather limited (i.e. 12 ppmv) effect on atmospheric CO2, certainly leaning towards the lower end of available estimates from the literature.*

*This contribution is certainly both stimulating and timely and will certainly be of interest to the climate science community. I have to say, however, that the conclusions are somewhat weakened by the reduced sensitivity of the model to increased Fe availability. As mentioned below (last point), I would urge the authors to reconsider the modern Fe budget, which would allow the argumentation to be more relevant and certainly more convincing.*

*I'm not a climate modeler and as such have mostly concentrated on commenting the paleoclimatic/biogeochemical aspects of the manuscript. My comments are listed below.*

*As far as I understand the model set up does not account for the T-dependency of the remineralization length scale.*

**General comment**

*As shown by Kwon et al., 2009 (NGeo), the most important parameter accounting for enhanced sequestration of CO2 into the ocean interior results from the redistribution of remineralized carbon from intermediate to bottom waters. In essence, the depth at which POC is being remineralized is not critical as long as POC respiration takes place at intermediate depths, from which nutrients and CO2 can rapidly be resupplied to the fertile surface ocean, with negligible consequences for atmospheric CO2 concentrations.*

*However, if the bulk of POC remineralization takes place in the deep ocean cell, then CO2 can be sequestered away from the atmosphere for centuries to millennia. So in essence, if the ballasting effect does not allow POC to be exported to the deep ocean, then one would expect the consequences for atmospheric pCO2 to be small.*

*I was wondering if you could come up with some sense on how generally colder temperatures characteristic of the LGM in combination with the ballasting effect would affect atmospheric CO2 concentrations. I understand that adding T-dependent POC remineralization rates would be computationally expensive. But this aspect should at least be discussed in some more details.*

As shown in Segschneider and Bendtsen (2013) for a HAMOCC global warming experiment, the inclusion of T-dependent remineralization has a more complex impact on the carbon sequestration than one would expect from a simple remineralization depth scale change (reduction for warming, increase for cooling). Compensating effects due to changes in remineralization and hence euphotic layer nutrient supply – driving changes in primary production – and further complication due to shifts in the ecosystem (opal vs. calcite producers) and resulting changes in surface alkalinity and hence $CO_2$-fluxes make it non-trivial to make any statements on the potential magnitude of including T-dependent remineralization on atmospheric $CO_2$. Segschneider and Bendtsen were planning to perform

corresponding experiments for a glacial ocean setup, but due to some unforeseen developments this has not materialized.

*Maybe you could also consider adding a few sentences regarding the role of dissolved O2 concentration on remineralization rates, since intermediate waters were probably better ventilated/oxygenated during the LGM (e.g. Jaccard and Galbraith, 2012 (NGeo); Galbraith and Jaccard, 2015 (QSR)).*

In the revised manuscript, effects of dissolved O2 concentrations on the simulated modern spatial transfer efficiency pattern are discussed (page 11, lines 21-26). We also highlight that ballasting effects can potentially modify the effects of changed oxygen concentrations during glacial cycles (page 18, lines 11-18).

**Detailed comment**

*p. 1, l. 13 – Köhler et al., 2017 do not present any ice-core CO2 data. Please remove.*

The 80ppm pCO$_2$ difference between the early Holocene and the LGM was estimated from the CO$_2$ data spline presented in Fig. 1a of Köhler et al. (2017). For that time period, the spline is based on data from the WAIS Divide Ice Core; we added the reference pointing directly to this data in the revised manuscript (Marcott et al. 2014).

*p. 2, l. 3 - . . . "enhanced aridity", is probably more adequate that "enhanced desert"*

We clarified: "… enhanced desert dust production and enhanced glacigenic dust production."

*p. 2, l. 3-4 - please add appropriate references*

We clarified that these are also results of the modelling studies referred to in the previous sentence (in particular, Mahowald et al., 2006).

*p. 2, l. 16 – please consider citing Hain et al., 2010 (GBC)*

Thanks, the reference was added to the list.

*p. 11, l. 24-25 – please note that this observation is consistent with paleoceanographic observations, which suggest enhanced export production in the South Atlantic during the LGM as a result of Fe-bearing dust fertilization (e.g. Kumar et al., 1995 (Nature), Martinez-Garcia et al., 2014 (Nature), Anderson et al., 2014 (Phil. Trans. R. Soc.)). Furthermore, using stable nitrogen isotopes as a proxy for the relative nitrate consumption by phytoplankton, Martinez-Garcia et al., 2014 (Nature) showed that the biological carbon pump was not only stronger but also more efficient, in line with the argument outlined here.*

Thank you for pointing the references out to us. We added the references to the introduction but decided that a comparison of the nitrate utilization changes reconstructed by Martinez-Garcia et al. to our results would bring us too far off track, also since the study is focused on iron fertilization effects and not on ballasting.

*p. 14, l. 8-10 - As mentioned above, there is ample evidence suggesting enhanced export production in the Subarctic Zone of the Southern Ocean as a result of Fe- fertilization (see reference above), including outside of the direct influence of the Patagonian dust plume (e.g. Lamy et al., 2014 (Science). I am somewhat surprised that the model is not able to reproduce the paleoceanographic evidence.*

Yes, we were also surprised and somewhat disappointed by that result (see response to your next comment). The disappointment turned into our motivation to fix this issue by using a more recent dust deposition field, but the implementation is still ongoing work.

*p. 15 – I'm a bit puzzled by the final remarks. In essence you imply that Fe concentrations are too high in your control run, in part to the shortcomings associated with the study published by Mahowald et al., 2006. As a consequence, adding Fe to simulate glacial conditions will not entail much of an effect on atmospheric CO2 concentrations. This certainly weakens the conclusions of the sensitivity study. Wouldn't it thus be possible to include model runs including the downscaled modern dust input?*

Understanding this may require a bit of a historical background: When starting our model development, we were not really aiming at an investigation of the iron fertilization effect on glacial $pCO_2$. Due to the standard model setup, however, in which dust is a source of iron, any change in the dust input intended to estimate the ballast effect on dust driven glacial $pCO_2$ decrease, will likewise have an effect on the amount of iron from the same dust input field. Therefore, we had to single out the effects of glacial dust on iron fertilization and enhanced settling velocities. And only then it turned out that the biological production was nowhere iron limited in the standard HAMOCC version. Likewise, we (both the authors of this study, and the model developers at MPI) were limited to the Mahowald et al. 2006 dust fields, as they were the only ones available with LGM/modern (and future) fields.

As discussed in our general response to all reviewers, we are currently working on the implementation of a more recent dust deposition reconstruction by Albani et al. (2016), which is expected to lead to iron limitation of phytoplankton growth rates in the simulated Southern Ocean, in line with modern observations. However, this development will take several months at least. And, because the lack of iron limitation occurs in both control simulations with and without ballasting and not only within the sensitivity runs, including the new dust field or using a dust field that is scaled down would require the repetition of the control simulations and of the sensitivity runs, and the release of a new standard version of HAMOCC, which we think is beyond the scope of this paper.

In the revised manuscript, we further highlight that the simulated iron fertilization effect is solely due to the fertilization of cyanobacteria growth and not comparable to the previous estimates from the literature, and we clarify that the focus of the manuscript is the description of the ballasting parameterization and the estimate of the LGM dust ballasting effect on atmospheric $CO_2$. To this end, we removed "and iron addition" from the manuscript title, modified the abstract (now explicitly mentioning diazotroph and non-diazotroph effects in the last sentence), and we explain already in the introduction that the iron effect is likely underestimated (and why we still do look at the iron effect; page 3 lines 6-12).

15    particle ballasting (Fig. 2d-e). The relatively high transfer efficiencies in the mid-latitudes in our simulation with ballasting are in line with high particle sinking speeds mostly due to ballasting by calcite particles (compare Fig. 2e to Fig. 3l).

Dust substantially affects sinking speeds and the transfer efficiencies in the Southern Ocean and South Atlantic off Patagonia, as well as in the Equatorial equatorial Atlantic due to Saharan dust deposition, in the North Atlantic, and in the Arctic Ocean. Note, however, that, in particular in the Arctic Ocean, where POC concentrations are low, even small quantities of dust can
20    lead to high sinking speeds, since our simple particle ballasting parameterisation only takes into account the density of the dust (in the limit of zero POC), and not its diameter. This leads to high sinking speeds in the model, while in reality, when POC concentrations are so low that particle collisions and aggregate formation are unlikely, the small and mostly individual dust particles sink very slowly according to Stoke's law. While this illustrates the a limitation of the simple particle ballasting parameterisation, the effect of the erroneously high sinking speeds in the Arctic on the carbon export is small: the integrated
25    carbon export at 90 m depth north of 80° N amounts to only 0.01 Gt C yr$^{-1}$, or just over 0.1 % of the global export of about 7 Gt C yr$^{-1}$.

In summary, although the simulated sinking speeds substantially differ between the simulations with and without particle ballasting, the biases of the setup with particle ballasting compared to observations remain similar to the biases of the standard setup. But the setup with particle ballasting now enables us to estimate the effect of glacial dust deposition on atmospheric
30    pCO$_2$.

**4 Sensitivity to LGM dust deposition**

**4.1 Experimental setup**

Starting from the simulation with particle ballasting described in the previous section (PI_BALLAST, with the opal dissolution rate reduced by a factor of 6, modern dust deposition, modern ocean circulation and pre-industrial GHGs), 3 experiments are performed to estimate the sensitivity of the ocean carbon cycle to the reconstructed LGM dust deposition. In all 3 sensitivity experiments, the prescribed monthly-mean modern dust deposition fields are replaced instantaneously by monthly-mean LGM dust deposition fields (see Fig. 5 for timeseries of the sensitivity experiments, and Fig. 6a-c for maps of the annual mean dust deposition rates and the LGM-modern dust deposition difference). In the first experiment, the LGM dust deposition rates are only used for the computation of dust concentrations in the water column, isolating the ballasting effect of the LGM dust deposition (LGM_BALLAST; solid blue lines in Fig. 5). In the second experiment, the LGM dust deposition rates are only used for the computation of the bio-available iron concentrations, isolating the iron fertilisation effect associated with the dust (orangeLGM_IRON; orange lines). In the third experiment, the LGM dust deposition is used for both, particle ballasting and iron fertilisation (pinkLGM_BOTH; pink lines).

**4.1 Atmospheric pCO$_2$ feedback**

To quantify the roles of particle ballasting and iron fertilisation by dust deposition changes for atmospheric pCO$_2$, the atmosphere–ocean CO$_2$ flux anomalies relative to the reference simulation with modern dust deposition (black lines in Fig. 5) are diagnosed at the end of each simulated year, and the prescribed atmospheric pCO$_2$ is updated accordingly. For an accumulated net atmosphere–ocean CO$_2$ flux anomaly of 2.1 Gt C into the ocean, the atmospheric CO$_2$ concentration is reduced by 1 ppmv. Note that only the surface ocean biogeochemistry responds to this atmospheric pCO$_2$ feedback. There is no effect on the atmospheric radiative transfer or climate; the prescribed climatological atmospheric forcing is unchanged.

If the atmospheric CO$_2$ concentration was not adjusted to the flux anomalies, a for example positive anomalous ocean CO$_2$ uptake would not result in a reduced atmospheric pCO$_2$, the atmosphere would behave like a CO$_2$ reservoir in our experiments, and the subsequent ocean CO$_2$ uptake would be overestimated. This is illustrated in a fourth experiment, in which LGM dust is prescribed for particle ballasting without applying the simple atmospheric pCO$_2$ box model described above (dashed blue line in Fig. 5a-b).

[revised manuscript text omitted]

---

## Referee Report (RR1)

Review of revised manuscript by Heinemann et al., for GMDD

I reviewed the manuscript previously and mainly commented on the appropriateness of the model's response to changing dust fluxes, the description of the ballasting parameterisation, and the inclusion of sediments. The authors have in my view addressed many of these comments appropriately. However, the authors have not been able to directly address the issue of a lack of iron limitation as this very understandably requires a significant amount of additional work to reparameterise and spin-up the model. The authors have been upfront about this and have added a lot of clear discussion about this in the manuscript.

Additional comments:

The effect of glacial iron fluxes on diazotrophs occurs in the tropical Pacific which is separate from the ballasting effect which occurs mainly in the high latitudes. It's unfortunate that the authors cannot directly address this, but more context/discussion could be given for the current results about the diazotrophs in the model, e.g., has this been quantified for LGM dust fluxes before?

The authors have now demonstrated there aren't any significant interactions between iron fertilisation and ballasting associated with the LGM dust forcing. But, any fertilisation effect in the Southern Ocean would be enhanced or diminished by the associated ballast effect which should be noted in the discussion. Additionally, any associated PIC:POC changes could be also different in the Southern Ocean.

The authors have added figures showing timeseries of organic carbon export, opal export and PIC:POC ratios. I am concerned that there are still significant trends in the ballasting simulations. The authors state that the reduced organic carbon export is due to nitrate depletion at the surface, which is fine, but what is causing this depletion? Figure 6 shows results from the first 100 years whilst Figure 5 shows deviations from these initial trends, which is somewhat confusing. I think this should at least be stated more clearly.

---

## Author Response (AR2)

**Response to Referee #1**

*My only comment would be to mention that you are talking about Figure 2c in the paragraph on page 12, lines 5-9.*

Agreed, a reference to Fig. 2c and 2f was added.

**Response to Referee #2**

*The authors put a lot of effort into addressing the reviewer comments but understandably aren't able to address one of the key issues that non-diazotroph phytoplankton are not iron limited anywhere in the model. However, there is no context or discussion of iron limitation of diazotrophs in the manuscript for the modern ocean or LGM ocean.*

Thanks for this comment. We added context with respect to the iron limitation of diazotrophs to the revised manuscript (please see response to the next comment).

*Additional comments:*
*The effect of glacial iron fluxes on diazotrophs occurs in the tropical Pacific which is separate from the ballasting effect which occurs mainly in the high latitudes. It's unfortunate that the authors cannot directly address this, but more context/discussion could be given for the current results about the diazotrophs in the model, e.g., has this been quantified for LGM dust fluxes before?*

Most of the N fixation in the present ocean is accomplished by only a few cyanobacteria species, the dominant of which is Trichodesmium (e.g., Falkowski et al., 1998). It has been illustrated that nitrogen fixation by Trichodesmium requires large amounts of iron, and that diazotrophy is thus limited by iron availability in wide areas of the present ocean (e.g., Falkowski et al. 1998, Berman-Frank et al. 2001, Kustka et al. 2003).

There is one (to our knowledge) previous study also suggesting that the effect of N fixation changes in the Pacific on atmospheric $CO_2$ in response to enhanced LGM dust deposition did contribute to the glacial $CO_2$ drawdown (Moore and Doney 2007). In their model – the Biogeochemical Elemental Cycling (BEC) ocean model with (apart from the dust fluxes) pre-industrial boundary conditions – N fixation changes combined with enhanced diatom production and export in response to the enhanced dust deposition led to an ocean $CO_2$ uptake equivalent to 16ppm.

We added this context to the Introduction (Section 1, page 2), and to the discussion of the iron fertilization results in Section 4.2 (page 16) and briefly in Section 5 (page 17).

*The authors have now demonstrated there aren't any significant interactions between iron fertilisation and ballasting associated with the LGM dust forcing. But, any fertilisation effect in the Southern Ocean would be enhanced or diminished by the associated ballast effect which should be noted in the discussion. Additionally, any associated PIC:POC changes could be also different in the Southern Ocean.*

The discussion of potential synergistic effects was extended accordingly, now mentioning this point explicitly (Section 5, page 18 in the document with tracked changes).

*The authors have added figures showing timeseries of organic carbon export, opal export and PIC:POC ratios. I am concerned that there are still significant trends in the ballasting simulations. The authors state that the reduced organic carbon export is due to nitrate depletion at the surface, which is fine, but what is causing this depletion?*

The nitrate depletion in the experiment with LGM dust for ballasting is caused by the increased aggregate sinking speeds (page 15 line 9 in the revised manuscript). The resulting faster export of nitrate is neither compensated by riverine influxes (organic matter weathering fluxes in the presented sensitivity experiments are prescribed and not altered), nor by nitrate fixation or denitrification changes.

*Figure 6 shows results from the first 100 years whilst Figure 5 shows deviations from these initial trends, which is somewhat confusing. I think this should at least be stated more clearly.*

The mentioned differences between the figures and panels are now highlighted in the introductory paragraph of the dust sensitivity results section (Section 4, page 12-13 in the document with tracked changes below).

[revised manuscript text omitted]

**Figure 5.** Effects of switching to LGM dust deposition for particle ballasting (LGM_BALL, blue), for the computation of iron fertilisation associated with the dust (LGM_IRON, orange), and for both (LGM_BOTH, pink) compared to the control simulation with ballasting and iron fertilisation based on modern dust deposition (PI_BALLAST, black). a, global mean sinking speed profiles; gray dashed line indicates the Martin-type sinking speed profile as used in the standard simulation without ballasting (PI_MARTIN); note that the black curve (PI_BALLAST) is hidden behind the orange curve (LGM_IRON), and that the blue curve (LGM_BALL) is hidden behind the pink curve (LGM_BOTH); b, global mean atmosphere–ocean $CO_2$ fluxes; positive values indicate a net flux from the atmosphere into the ocean; solid lines are 50-year running means; the dots show annual means of the $CO_2$ fluxes in PI_MARTIN to illustrate the large interannual variability (e.g., due to ocean circulation variability) compared to the dust-induced anomalies; the blue dashed line is the $CO_2$ flux in a simulation with LGM dust for ballasting where the atmospheric $pCO_2$ was *not* updated every year according to the atmosphere–ocean $CO_2$ flux anomalies of the previous year; c, atmospheric $pCO_2$ change based on the annual mean atmosphere–ocean $CO_2$ flux anomalies relative to PI_BALLAST; the gray dashed line shows the sum of the LGM_BALL and LGM_IRON $pCO_2$ change; d, global organic carbon export anomaly relative to PI_BALLAST in 90 m depth; e, global opal export anomaly relative to PI_BALLAST in 90 m depth; f, anomaly of the global ratio of particulate inorganic carbon (PIC, calcite) to particulate organic carbon (POC) within the particle export in 90 m depth.

Fig. 5 show the longer-term evolution of the induced anomalies over the entire 6500 years of the sensitivity experiments. The maps shown in Fig. 6d-l are computed from the first 100 years of the sensitivity experiments to capture the time of the largest atmosphere–ocean $CO_2$ flux anomaly signal.

**4.1 Role of LGM dust as ballast**

[revised manuscript text omitted]